# Molecular diet analysis enables detection of diatom and cyanobacteria DNA in the gut of *Macoma balthica*

**Julie A. Garrison**[1]*, **Nisha H. Motwani**[2,3], **Elias Broman**[1,4], **Francisco J. A. Nascimento**[1,4]

**1** Department of Ecology, Environment and Plant Sciences, Stockholm University, Stockholm, Sweden, **2** School of Natural Sciences, Technology and Environmental Studies, Södertörn University, Huddinge, Sweden, **3** Department of Environmental Science, Stockholm University, Stockholm, Sweden, **4** Baltic Sea Centre, Stockholm University, Stockholm, Sweden

* julie.garrison@su.se

**Data Availability Statement:** Data are available as Supporting Information and sequences are available in the NCBI sequence read archive (SRA) database (http://www.ncbi.nlm.nih.gov/sra) under

## Abstract

Detritivores are essential to nutrient cycling, but are often neglected in trophic networks, due to difficulties with determining their diet. DNA analysis of gut contents shows promise of trophic link discrimination, but many unknown factors limit its usefulness. For example, DNA can be rapidly broken down, especially by digestion processes, and DNA provides only a snapshot of the gut contents at a specific time. Few studies have been performed on the length of time that prey DNA can be detected in consumer guts, and none so far using benthic detritivores. Eutrophication, along with climate change, is altering the phytoplankton communities in aquatic ecosystems, on which benthic detritivores in aphotic soft sediments depend. Nutrient-poor cyanobacteria blooms are increasing in frequency, duration, and magnitude in many water bodies, while nutrient-rich diatom spring blooms are shrinking in duration and magnitude, creating potential changes in diet of benthic detritivores. We performed an experiment to identify the taxonomy and quantify the abundance of phytoplankton DNA fragments on bivalve gut contents, and how long these fragments can be detected after consumption in the Baltic Sea clam *Macoma balthica*. Two common species of phytoplankton (the cyanobacteria *Nodularia spumigena* or the diatom *Skeletonema marinoi*) were fed to *M. balthica* from two regions (from the northern and southern Stockholm archipelago). After removing the food source, *M. balthica* gut contents were sampled every 24 hours for seven days to determine the number of 23S rRNA phytoplankton DNA copies and when the phytoplankton DNA could no longer be detected by quantitative PCR. We found no differences in diatom 18S rRNA gene fragments of the clams by region, but the southern clams showed significantly more cyanobacteria 16S rRNA gene fragments in their guts than the northern clams. Interestingly, the cyanobacteria and diatom DNA fragments were still detectable by qPCR in the guts of *M. balthica* one week after removal from its food source. However, DNA metabarcoding of the 23S rRNA phytoplankton gene found in the clam guts showed that added food (i.e. *N. spumigena* and *S. marinoi*) did not make up a majority of the detected diet. Our results suggest that these detritivorous clams therefore do not react as quickly as previously thought to fresh organic matter inputs, with other phytoplankton than

project accession number PRJNA821398 (see S1 table for additional information).

**Funding:** Funding was provided by the Swedish Environmental Protection Agency's Research Grant (NV-802-0151-18) to FJAN in collaboration with the Swedish Agency for Marine and Water Management and by the Foundation for Baltic and East European Studies, Project no. 56/19 to NHM The funders had no role in study design, data collection and analysis, decision to publish, or preparation of the manuscript.

**Competing interests:** No authors have competing interests, financial or otherwise.

large diatoms and cyanobacteria constituting the majority of their diet. This experiment demonstrates the viability of using molecular methods to determine feeding of detritivores, but further studies investigating how prey DNA signals can change over time in benthic detritivores will be needed before this method can be widely applicable to both models of ecological functions and conservation policy.

## Introduction

Detritivores are essential to nutrient recycling and process more organic matter than primary consumers [1]. Despite this importance, they have been historically understudied in terms of food webs, as they are logistically difficult to place within the web [2] and present many methodological difficulties in diet determination and interpretation, as they do not fit into one trophic level [3]. This is particularly true for benthic detritivores that are important for secondary production in the second largest habitat on Earth–the soft sediment of aquatic ecosystems [4,5]. Determining the diet of benthic detritivores with good taxonomic resolution is therefore vital for an improved understanding of marine food webs and the cycling of nutrients happening in marine soft sediments [1,6,7].

Much of the knowledge about marine detritivores trophic links comes from stable isotope techniques. While stable isotope analysis is relatively easy to apply, it presents several drawbacks: it is often difficult to get accurate food source signals due to the heterogeneity of detritus [8], the signals are difficult to interpret [3], and it measures what the organism has incorporated into the tissues, relying on a period to empty the digestive tract before sampling the stable isotopes [9–12]. Some of these concerns are alleviated by higher resolution isotopic methods, such as compound-specific stable isotope analysis that shows promising results for interpretation of diet [13], but benthic detritivores seem to present problems by remaining stable despite changing food sources [14].

Molecular tools have been important to the recent advances in food web understandings [15]. They allow for identification of prey species in predators that are not readily observable due to sampling difficulties [16], such as soft bodied prey that cannot be identified in the guts due to faster digestion [17], or cryptic prey species [15]. There have been hurdles to applying molecular methods to food web ecology, namely: developing databases of reference sequences [18]; bias in each step (DNA extraction, purification, and PCR amplification; [19]; and quantification of prey [20]). Real-time quantitative polymerase chain reaction (qPCR) allows for both quantification and, with some previous diet knowledge, identification of the prey [21,22]. However, the length of time that one can detect prey DNA after consumption has been shown to vary by predator species [23]. This is important for understanding food web energetics, and for any future studies that use molecular methods to determine diet, as the length of time prey DNA is detectable has been shown to depend on: predator species identity [24,25]; prey species identity [26,27]; temperature [28,29]; and meal size [30,31]. Other studies have found that prey DNA is detectable in predators from only a few hours to several weeks [32,33]. qPCR has increased our knowledge of diet and trophic links in terrestrial invertebrates [24,27,34–38], aquatic vertebrates [21,29,39,40] and pelagic invertebrates [32,41–45]. However, studies of benthic invertebrate diet using qPCR are lacking.

The Baltic Sea is one of the largest brackish water systems, containing a reduced macrofauna (invertebrates larger than 1 mm) species richness due to the sharp salinity gradient and young geological age [46,47]. Below the photic zone, only a few detritivorous macrofauna

species are present; one of the most important is the Baltic clam *Macoma* (*Limecola*) *balthica* (Linnaeus, 1758). Since the 1970s, *M. balthica* has been determined to be one of the most important species in the Baltic Sea in terms of ecosystem structure and function [48]. According to [49], *M. balthica* is responsible for mineralization of over a third of sedimented phytoplankton in the Baltic Proper, but also increases greenhouse gas (methane) emissions from sediment activity [50]. *M. balthica* are also important prey of commercially important benthic fish, such as *Platichthys flesus* [51]. *M. balthica* is a facultative suspension and surface deposit feeder that is thought to consumes detritus (including bacteria), phytoplankton, and possibly meiofauna [52]. Most of benthic secondary production growth in the Baltic Sea is coupled to the settling of the spring bloom (typically April in the study area) while relying on detritus for most of the year [49,52–55]. Due to warmer temperatures, a longer growing season, and nutrient enrichment, the cyanobacteria-dominated summer phytoplankton bloom (typically July-August in the study area) has been increasing in magnitude and duration in the Baltic Sea [56,57]. In contrast, the diatom-dominated spring phytoplankton bloom has been shrinking in volume and duration [57–59]. It is well known that cyanobacteria lack the essential fatty acids found in diatoms [60], and potentially contain harmful cyanotoxins [61], and are thus considered to be of lower nutritional status [62]. This change in phytoplankton timing and nutritional status can have consequences on the benthic invertebrate community, particularly deposit feeders [49]. *M. balthica* is known to consume both diatoms and cyanobacteria, but appears to prefer the former over the latter [63,64]. Additionally, *M. balthica* is a facultative suspension feeder [65] that has been shown to have high feeding plasticity that is dependent on abiotic factors, namely physical forcing such as winds and waves [66]. Indeed, [66] found that *M. balthica* relied to a larger extent on sediment-bound organic material in sediments in protected bays, but increased their feeding on suspended particulate organic material in exposed sites. DNA-based methodologies could help elucidate feeding preferences and individual trophic plasticity (i.e. changes in feeding strategies and food selection). The application of qPCR techniques to quantify DNA fragments of these two phytoplankton groups has the potential to provide valuable data regarding this question, and DNA metabarcoding of the entire phytoplankton community helps place these targeted groups into the larger diet context. Empirical quantitative data on feeding magnitude on these two phytoplankton groups is critical to predict the consequences of these potential changes for *M. balthica* dynamics and benthic secondary production in the Baltic.

With these knowledge gaps in mind, we performed an experiment where two groups of the Baltic clam *M. balthica* from differing levels of physical forcing exposure (sheltered or exposed) were fed two common species of phytoplankton that are known to be part of their natural diet: the cyanobacteria *Nodularia spumigena* and the diatom *Skeletonema marinoi* [64]. First, we aimed to confirm the presence of phytoplankton DNA in the digestive tract of *M. balthica*, both through qPCR of *N. spumigena* and *S. marinoi*, and through DNA metabarcoding targeting the whole phytoplankton communities. Secondly, we aimed to quantify using qPCR: i) the length of time that *N. spumigena* and *S. marinoi* DNA can be detected in *M. balthica* guts, ii) if feeding by the two *M. balthica* regional groups on each of the phytoplankton species differs, which would likely be due to trophic plasticity and differing availability of phytoplankton on their feeding strategy. We predicted that both phytoplankton species would be consumed by all clams, but that *S. marinoi* would be consumed at a higher magnitude due to the higher nutritional value, and that the sheltered southern clams would have higher phytoplankton consumption due to higher deposit feeding, where the phytoplankton in this experiment could be found. Based on previous studies of similar organisms, we expected the phytoplankton signal to be lost after 3–5 days [25,53].

## Materials and methods

### Sampling and experimental set-up

The Baltic clam *M. balthica* were sampled from two regions in the Baltic Sea and utilized in the experiment (S1 Fig). Benthic sled drags were used to sample in the southern Stockholm archipelago outside Askö marine research station in Sweden on 20 October 2017 (Fig 1; station S; samples pooled from 2 stations with center 58.83 N, 17.55 E, and maximum distance between stations 1.76 km, average 34 m water depth, range 28–39 m). The sediment was sieved through 0.5 mm mesh, and the *M. balthica* were kept in oxygenated water with sediment from the station at constant 4˚C in the dark to simulate field conditions at Stockholm University until the start of the experiment. *M. balthica* were also sampled with benthic sled drags from the northern Stockholm archipelago, outside of Norrtälje, Sweden, on 18 January 2018 (station N; samples pooled from 5 stations with center 59.54 N, 19.43 E, and maximum distance between stations 24 km, average 50 m water depth, range 37–62 m). Briefly, multiple stations were pooled within each area to counteract any local irregularities in bottom composition or sedimentation rate while still preserving the regional trends in physical characteristics to ensure that we were investigating regional level effects, and not individual diet. *M. balthica* density is

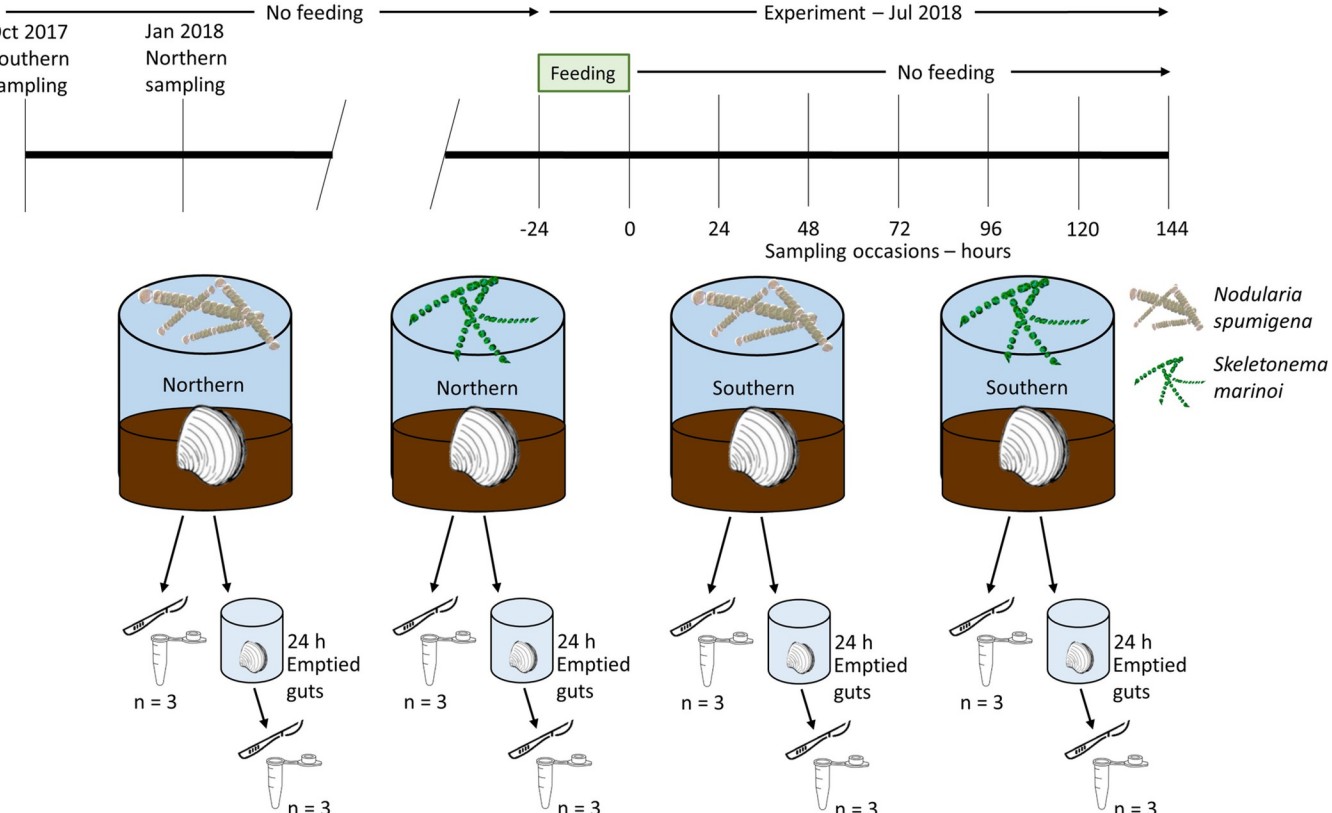

**Fig 1. Experimental timeline and setup.** Timeline (top) and experimental setup (bottom). Northern *Macoma balthica* samples were taken from the northern Stockholm archipelago, while southern samples were taken from the southern Stockholm archipelago. Each regional group was fed either the cyanobacteria *Nodularia spumigena* (brown) or the diatom *Skeletonema marinoi* (green), to create four treatments, and allowed to feed for 24 hours before the food source was removed. Samples were taken every 24 hours for one week, and either dissected and frozen immediately (left samples, represented by scalpel and tube) or allowed to empty their guts for 24 hours in freshwater before dissection and freezing (right; see S1 File for more information). All clams were kept in their original sediment without feeding for at least six months at the time of experiment start.

lower in the northern stations, and thus more stations were needed to obtain enough individuals. The southern station is considered more sheltered, as it is located further into the archipelago, while the northern station is more exposed to the open sea. These samples were also sieved through 0.5 mm mesh and kept in oxygenated water and sediment from the stations at constant 4˚C in the dark to simulate field conditions at Stockholm University until the start of the experiment. Fresh water was replenished as needed in both groups due to evaporation and to maintain salinity around natural values (6.7 PSU), and the clams were without new food sources for about six months before the start of the experiment.

Two food sources were utilized in the experiment, grown in the lab as cultures: the cyanobacteria *N. spumigena* (Ns; strain K-1537, provided by the Norwegian Institute for Water Research, Norway) and the diatom *S. marinoi* (Sm; strain LYS6AAF, provided by the Department of Environmental Science and Analytical Chemistry, Stockholm University). See [67] for more information on the phytoplankton cultures. Both species are common in the Baltic Sea, with *S. marinoi* present in the spring diatom bloom, and *N. spumigena* in the summer cyanobacterial bloom.

The experiment was run at Stockholm University, started on 9 July 2018, and final samples taken on 16 July 2018. A fully crossed factorial design was used with the two regional groups of *M. balthica* and two species of phytoplankton to create four treatments: northern clams with *N. spumigena*, northern clams with *S. marinoi*, southern clams with *N. spumigena*, and southern clams with *S. marinoi*. At the start of the experiment, 50 mL of each phytoplankton culture was mixed with sediment from the two stations (approximately $3.3 \times 10^{-3}$ g C m$^{-2}$ *N. spumigena* or $5.5 \times 10^{-5}$ g C m$^{-2}$ *S. marinoi*, comparable to sedimentation rates in the study area [68]) in four clear cylindrical plexiglass mesocosms (18.5 cm height, 13 cm diameter) with sediment to a depth of 4 cm and brackish water (6 PSU). Forty-five *M. balthica* were added to each of the four treatments (3383 ind m$^{-2}$, comparable to natural densities in the Stockholm archipelago; [69,70]) with oxygenation and allowed to feed for 24 hours. *M. balthica* have been shown to feed down to a depth of 6 cm [71], so the food was available for consumption. At the same time that clams were added to the fresh phytoplankton amended sediments, clams from each region not fed with the added fresh phytoplankton were sampled to serve as baseline measurements (-24 h samples, n = 3). After 24 hours in the fresh phytoplankton amended sediment, the clams were removed from the food sediment and placed into four separate clear cylindrical plexiglass mesocosms (23 cm height, 17 cm diameter) with their original sediment (no added food) to a height of 3 cm and brackish water with oxygenation. Six replicate clams were sampled from each treatment at 0, 24, 48, 72, 96, 120, and 144 hours. Half of the sampled clams (n = 3) were dissected for their guts immediately and frozen individually, while the other half (n = 3) were placed individually in 25 mL fresh water for 24 additional hours to empty their guts before being dissected for their guts and frozen individually. See Fig 1 for experimental setup, and S1 File and S4 Fig for emptied gut *M. balthica* results and discussion. The empty guts treatments are intended to also serve as negative controls, as 24 hours without food is commonly used with the assumption that macrofauna will completely empty their guts during these conditions [10]. Dissections took place using sterilized (10% bleach then rinsed with MilliQ water) forceps and blade on a sterilized (10% bleach then 70% ethanol) petri dish. Three replicates from each region were sampled before feeding (-24 h) to act as a baseline, and three replicates per treatment immediately after the 24 hour feeding window (0 h) to act as a positive control. Any dead *M. balthica* found were removed from the analysis. For *M. balthica* wet weight, please see S1 Table. While individual shell lengths were not measured, all clams were between 10–20 mm to reduce variation by life stage and particle size processing limitations while feeding.

## DNA extraction and qPCR

Total DNA was extracted from the dissected *M. balthica* guts using a modified DNeasy Blood & Tissue kit (Qiagen), using a 3 hour incubation at 56˚C and then bead beating with 0.5 mm glass beads at 6.5 m s$^{-1}$ for 60 seconds using a FastPrep 24 (MP Bio). DNA was cleaned with DNeasy PowerClean Pro Cleanup kit (Qiagen). DNA was checked for concentration and purity on a NanoDrop One UV-vis spectrophotometer (Thermo Scientific) before and after cleaning. All samples were diluted with sterile molecular-grade water (MilliporeSigma Ultra-pure Water for Molecular Biology, Fisher Scientific) to approximately 5 ng μL$^{-1}$ DNA before qPCR.

SYBR Green qPCR assays with primer targets for *N. spumigena* and *S. marinoi* were used to detect respective phytoplankton in northern and southern treatments. For each phytoplankton species, the same culture fed to the clams was DNA extracted and diluted to make five serial dilutions that were quantified alongside the experimental samples to create a standard curve for determination of phytoplankton DNA concentrations. The qPCR targeting *N. spumigena* used the forward primer NTS (5'-TGTGATGCAAATCTCAMA-3'; [72]) and reverse primer 1494R (5'-TACGGCTACCTTGTTACGAC-3'; [73]) to amplify a 200 bp segment of the 16S rRNA gene region. Each reaction mix was set up as follows: 10 μL of 1 × SYBR green (BIO-RAD), 1 μL each of 10 μM primers, 5 μL sterile molecular-grade water (MilliporeSigma Ultrapure Water for Molecular Biology, Fisher Scientific), and 3 μL DNA template. qPCR conditions were run using a StepOne Real-Time PCR cycler (Applied Biosystems) as follows: initial denaturing of 15 minutes at 95˚C, followed by 40 cycles of 30 seconds at 94˚C, 30 seconds at 53˚C, and 30 seconds at 72˚C, followed by a final elongation hold of 10 minutes at 72˚C, and finally a melt curve analysis of 5 seconds at 65˚C increasing 0.5˚C per cycle to 95˚C [44]. The *S. marinoi* qPCR used the forward primer 18SF-Diatom-487 (5'-GGTCTGGCAATTGGAA TGAGAAC-3') and reverse primer 18SR-Diatom-615 (5'-CTGCCAGAAATCCAACTACGAG-3') to amplify a 128 bp segment of the 18S rRNA gene V3 region [74]. Molarity of the reaction mix was as follows: 10 μL of 1 × SYBR green (BIO-RAD), 1 μL each of 10 μM primers, 5 μL sterile molecular-grade water (MilliporeSigma Ultrapure Water for Molecular Biology, Fisher Scientific), and 3 μL DNA template. Conditions for the qPCR were run using a StepOne Real-Time PCR cycler (Applied Biosystems) as follows: initial denaturing of 3 minutes at 98˚C, followed by 40 cycles of 5 seconds at 98˚C, and 45 seconds at 62˚C, followed by a melt curve analysis of 5 seconds at 62˚C increasing 0.5˚C per cycle to 95˚C (modified protocol from [74] with recommendations from BioRad for SYBR Green using Applied Biosystems StepOne).

## Sequencing and bioinformatics

Samples for sequencing were selected to allow for a more comprehensive analysis of gut contents over the time points in all treatments (for selected samples, please see S1 and S2 Tables). DNA metabarcoding was conducted targeting domain V of the 23S rRNA gene, with the forward primer P23SrV_f1 (5'- GGACAGAAAGACCCTATGAA -3') and reverse primer P23SrV_r1 (5'- TCAGCCTGTTATCCCTAGAG -3') to amplify a 410 bp amplicon [75]. Conditions for the PCR were run as follows: initial denaturing of 94˚C for 2 min, followed by 35 cycles of 94˚C for 20 s, 55˚C for 30 s, and 72˚C for 30 s, and finally 72˚C for 10 min. Sequencing by synthesis was then performed on a NovaSeq 6000 (Illumina) S Prime lane with a 2 × 250 bp paired-end setup. Library preparation and sequencing were carried out by NovoGene Ltd.

The raw sequence data was analyzed following in R (v4.1.1 [76]) using the DADA2 R package version 1.21.0 [77]. Parameters used included, for the quality trimming: truncLen = c (215,215, maxEE = 2, truncQ = 2, maxN = 0, and rm.phix = TRUE; error model using nbases = 1e8; merging read pairs with minOverlap = 10, maxMismatch = 0, and the chimera

removal step with method = "consensus". The amplicon sequence variants (ASV) were annotated using the µgreen-db r1.1 database [78], with addition of 33 *N. spumigena* and 4 *S. marinoi* annotations from the National Center for Biotechnology Information (NCBI) Basic Local Alignment Search Tool (BLAST) [79]. This database was chosen due to its curation, which helps ensure integrity of sequences and taxonomy, and specificity of the 23S rRNA gene [78]. Singletons were removed, and any ASV that could not be assigned at the Kingdom level in the phyloseq package version 1.38.0 [80].

## Data and statistical analysis

Values for qPCR were converted from StepOne Real-Time PCR cycler threshold cycle ($C_t$) values to number of copies using linear regression equations from serial dilutions of phytoplankton culture standard concentrations (for $C_t$ and number of copies, see S1 Table). The number of copies were then multiplied by the dilution factor (when diluting to 5 ng µL$^{-1}$ before qPCR) to get copy number µL$^{-1}$. All statistical tests were performed in R (v.4.1.2, [76]). To test for differences in feeding on the two phytoplankton species, linear regressions were plotted to the copy number µL$^{-1}$ over time separately for *N. spumigena* and *S. marinoi* and the slopes compared using the lm function in the stats package version 4.1.2 [76]. Differences between regions, sampling time points, and "emptied" and "non-emptied" guts (results presented in S1 File and S4 Fig) within each phytoplankton species were tested with a three-way analysis of variance (ANOVA; aov function in stats package) allowing for interaction after checking assumptions of normality with a Shapiro-Wilk test (shapiro.test function in stats package) and homogeneity of variance with a Levene test (leveneTest function in car package version 3.0.12; [81]). The *S. marinoi* dataset was log transformed, and the *N. spumigena* dataset was square root transformed to meet assumptions, and subsequent statistics were run on transformed data. A Tukey post-hoc test was utilized in the case of significant differences with the TukeyHSD function in the stats package. Multiple linear regression models were run to compare differences in slopes and intercepts between regions by phytoplankton species consumed, and between full and emptied guts (results presented in S1 File and S4 Fig). Again, the log-transformed *S. marinoi* and square root transformed *N. spumigena* datasets were used to meet the assumption of normal distribution. For the sequencing data, the ASVs assigned taxonomy were normalized as relative abundances (%) and differences between regions and fed phytoplankton were analyzed using non-metric multidimensional scaling (NMDS) based on the Bray-Curtis dissimilarity index using the ordinate function within the phyloseq package. Additionally, permutational multivariate analysis of variance (PERMANOVA) using the adonis function of the vegan package version 2.5.7 [82] was utilized to test for significant differences between regions and phytoplankton. Links between phytoplankton and *M. balthica* were visualized with the ChordDiagram function in the circlize package version 0.4.14 [83].

## Results

### Time of detection

DNA of phytoplankton was still detectable in the guts of *M. balthica* after 144 hours (one week) with qPCR (Fig 2), our longest time period after removal from the sediment with fresh phytoplankton. Additionally, there were no significant differences in phytoplankton DNA concentrations detected between sampling time points for either *S. marinoi* or *N. spumigena* (Table 1), indicating no significant increase or decrease in phytoplankton DNA detected over time. There were no significant linear trends, with the exception of an unexpected increased in detection of *S. marinoi* DNA in the southern clams over time (Table 2).

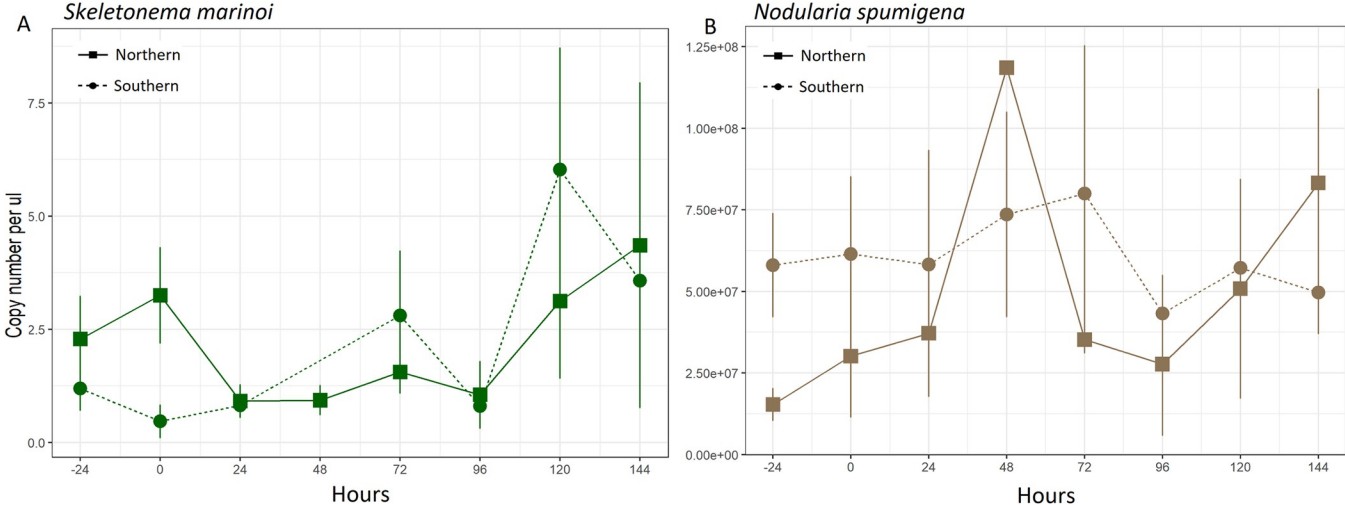

**Fig 2. Phytoplankton consumed by *Macoma balthica* determined by qPCR assays.** Phytoplankton consumption by *M. balthica*, determined by qPCR and reported in number of copies μL$^{-1}$ (16S rRNA gene for *Nodularia spumigena* and 18S rRNA gene for *Skeletonema marinoi*). Panel A represents consumption of the diatom *S. marinoi* (green), while B represents the cyanobacteria *N. spumigena* (brown). Squares and solid lines represent the northern Stockholm archipelago clams, and circles and dotted lines represent the southern Stockholm archipelago clams. Error bars represent standard error.

## Feeding on phytoplankton

The *M. balthica* from different regions showed no difference in abundance of *S. marinoi* 18S rRNA gene fragments in the gut (Fig 2A; Table 1), suggesting similar magnitudes of feeding on this phytoplankton species. However, the southern clams contained significantly higher ($F_{1,50}$ = 23.5, p < 0.001) amounts of *N. spumigena* 16S rRNA gene fragments in their guts than the northern clams (Fig 2B; Table 1), but the southern clams showed already higher baseline values at -24 h (Fig 2B).

## Diet determined by metabarcoding

We sequenced a total of 1,831,868 reads within 5,172 ASVs. We detected phytoplankton 23S rRNA amplicons in all samples sequenced (Fig 3, S1 Table). After filtering, our dataset consisted of 1,629,859 reads and 219 ASVs found in *M. balthica* digestive tracts, but 122 ASVs could not be assigned to the Kingdom level (56%) and are thus not included in figures, but were included in the statistical analysis. Cyanobacteria, Chlorophyta, and Ochrophyta were the phyla with the most ASVs, respectively, across all samples. In terms of class taxonomy, Cyanophyceae made up the most ASVs, followed by Eustigmatophyceae (Ochrophyta),

**Table 1. Differences in phytoplankton detection in *Macoma balthica* by qPCR.**

| Response variable | Independent variable | SS | df | F | p |
|---|---|---|---|---|---|
| *Skeletonema marinoi* number of copies μL$^{-1}$ | Time | 1.4 | 7 | 0.78 | 0.61 |
| | Region | 0.41 | 1 | 1.61 | 0.21 |
| *Nodularia spumigena* number of copies μL$^{-1}$ | Time | $3.3 \times 10^7$ | 7 | 0.64 | 0.72 |
| | Region | $1.7 \times 10^8$ | 1 | 23.5 | < 0.001 |

Analysis of variance (ANOVA) results, using time point (-24, 0, 24, 48, 72, 96, 120, or 144 hours) and region of *M. balthica* (northern or southern Stockholm archipelago) as factors to predict number of copies μL$^{-1}$ found in *M. balthica* guts of two phytoplankton prey, the diatom *S. marinoi* and the cyanobacteria *N. spumigena*, determined by qPCR.

**Table 2. Multiple linear regression of phytoplankton detection in *Macoma balthica* by qPCR over time.**

| | *Skeletonema marinoi* | | *Nodularia spumigena* | |
|---|---|---|---|---|
| | **Southern *M. balthica* region** | **Northern *M. balthica* region** | **Southern *M. balthica* region** | **Northern *M. balthica* region** |
| (Intercept) | -0.05 | 0.09 | 8092.53 *** | 5124.14 *** |
| | (0.08) | (0.08) | (453.59) | (487.77) |
| Time point | 0.18 * | -0.01 | -267.01 | 642.57 |
| | (0.08) | (0.08) | (459.68) | (493.98) |
| N | 36 | 42 | 38 | 40 |
| Adj. R$^2$ | 0.1 | -0.02 | -0.01 | 0.02 |

Multiple linear regression model outputs for detecting linear trends in phytoplankton number of copies $\mu L^{-1}$ found in *M. balthica* guts for the diatom *S. marinoi* and the cyanobacteria *N. spumigena* in the southern or northern Stockholm archipelago *M. balthica* clams. All continuous predictors are mean-centered and scaled by 1 standard deviation.

*** $p < 0.001$

** $p < 0.01$

* $p < 0.05$.

Trebouxiophyceae (Chlorophyta), and Mamiellophyceae (Chlorophyta). Somewhat surprisingly, Synechococcales made up the most ASVs of Cyanophyceae, followed by Nostocales, the order to which *N. spumigena* belongs. Trebouxiophyceae and Mamiellophyceae made up the most ASVs of Chlorophyta, and Ochrophyta was dominated by one single species, *Nannochloropsis gaditana. S. marinoi* belongs to the Bacillariophyceae class within the Ochrophyta phylum, which was not detected in our top ASVs. However, we found diatoms (order Chaetocerotales) and cyanobacteria (order Nostocales among others) in samples from all time points. Relative abundance at the species level for all samples can be found in S2 Fig. We found no significant difference between initial *M. balthica* before our experiment for our two feeding treatments (Fig 4; PERMANOVA; $F_{1,19} = 0.84$, $p = 0.64$) or the interaction of phytoplankton and region ($F_{1,19} = 0.82$, $p = 0.67$), but a significant difference by *M. balthica* region (Fig 4; $F_{1,19} = 1.77$, $p = 0.043$).

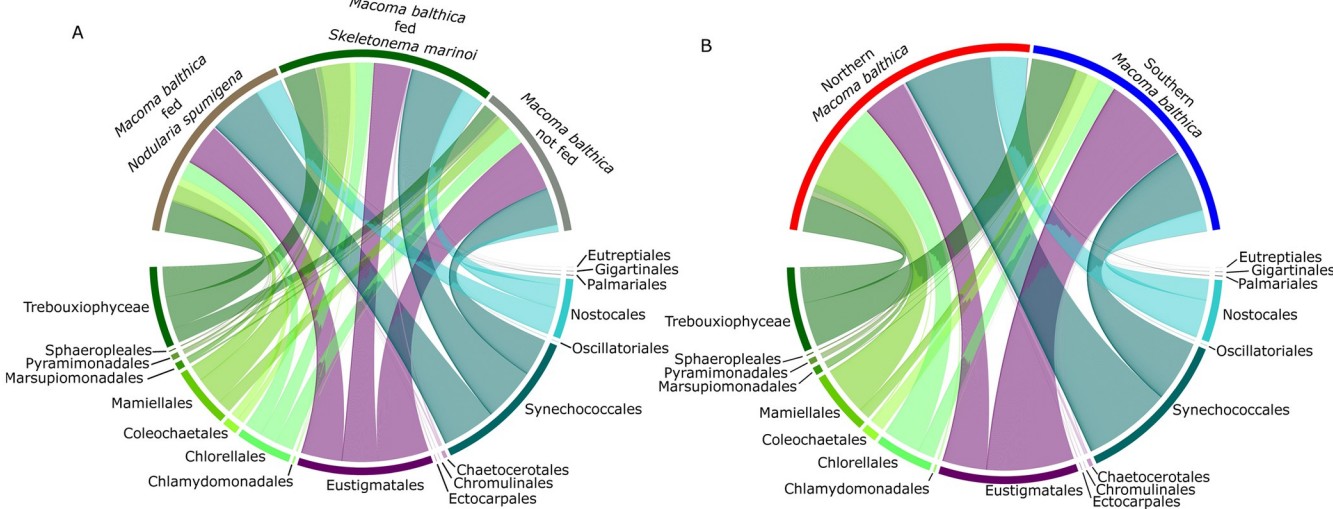

**Fig 3. Phytoplankton consumption by *Macoma balthica*.** The main orders of phytoplankton detected in the digestive tract of *M. balthica*, from selected samples sequenced after 23S rRNA gene metabarcoding, divided by phytoplankton fed (A; cyanobacteria *Nodularia spumigena* (brown), diatom *Skeletonema marinoi* (green), or not fed (gray)) and region (B; northern (red) and southern (blue) Stockholm archipelago). The size of the colored lines denotes the relative abundance (%) of the various taxa detected in the digestive tract of *M. balthica*.

## Phytoplankton

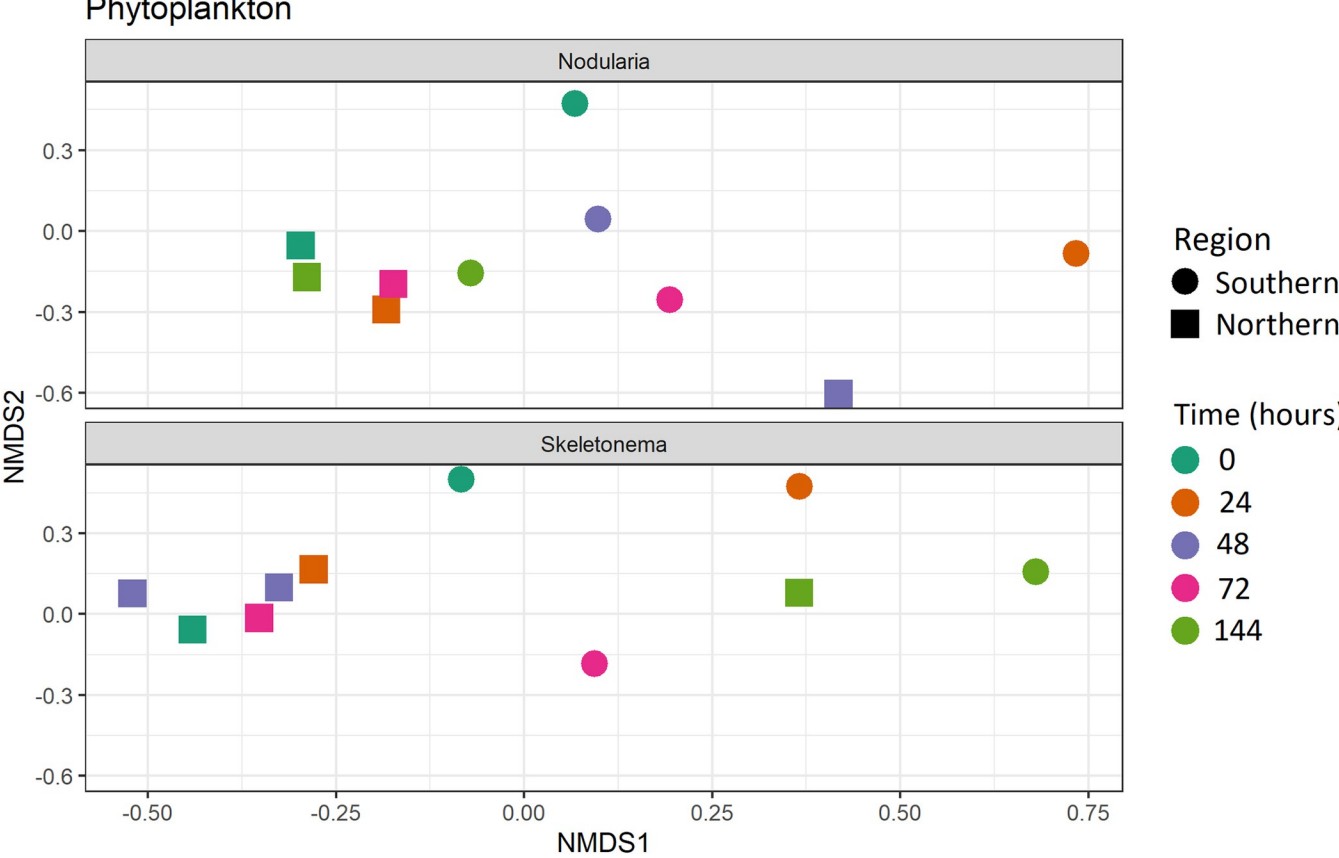

**Fig 4. Phytoplankton community consumed by *Macoma balthica* ordination.** Non-metric multidimensional (NMDS) scaling based on the Bray-Curtis dissimilarity index of DNA 23S rRNA gene metabarcoding communities in guts of *M. balthica* found by phytoplankton fed (cyanobacteria *Nodularia spumigena* on top, diatom *Skeletonema marinoi* on bottom), with shapes showing region (circle southern Stockholm archipelago, square northern Stockholm archipelago), and sampling time point (stress = 0.16). For full NMDS see S3 Fig.

## Discussion

We could detect DNA fragments of both the diatom *S. marinoi* and the cyanobacteria *N. spumigena* in the guts of *M. balthica* for the whole duration of our experiment (1 week) using real-time quantitative PCR (qPCR), refuting our hypothesis that the phytoplankton DNA signal would be lost after a few days. The two regional groups fed on *S. marinoi* at a similar magnitude, but the southern Stockholm archipelago clams showed higher *N. spumigena* DNA detection than the northern Stockholm archipelago clams. Surprisingly, the two species of phytoplankton that were fed to *M. balthica* were not among the most detected sequences in the metabarcoding dataset, though we did find diatoms and cyanobacteria in the guts at all time points, confirming the long gut residence time detected by qPCR. This shows that species-specific qPCR is a sensitive method that can detect DNA of prey in the guts, although it may not make up a large portion of the overall diet. This could be a result of too little time to react to the fresh organic matter, which was 24 hours in our study, and perhaps varied access time to fresh food should be evaluated in future experiments. Additionally, we found that phytoplankton DNA in sediments can remain viable and ingested by benthic detritivores for at least six months following organic matter input, which can have important implications for the understanding of benthic detritivore feeding ecology.

Unlike previous studies that looked at the prevalence of prey DNA in e.g. insect and spider consumers [25,29,33,35–38], we found no consistent increase in phytoplankton detection or linear decrease in the guts of the clam *M. balthica* following feeding. Additionally, we likely required more sampling time points such as sooner after feeding (within 30 minutes) to detect initial increases in phytoplankton DNA [32], as well as longer than one week after feeding to capture the decrease in phytoplankton DNA detection [33]. For example, [33] still detected prey DNA in spiders after 14 days, while [36] and [38] did not detect prey DNA in spiders after 5 days. As our study is the first investigation into benthic detritivore feeding by qPCR that we are aware of, and most work involving time of prey detection (from a few hours to 14 days) has been performed on terrestrial invertebrates (insects and spiders; [23]), the gut residence time detection in benthic invertebrates will require further calibration. Additionally, studies focusing on meal size [30] and temperature [29] effect on prey DNA detection should be a priority, while increasing sample size for adequate replication.

## Feeding of the Baltic clam on two phytoplankton species

Here, we were able to investigate differences in consumer-related factors by using two regions of *M. balthica*, and prey-related factors by using two species of phytoplankton prey. Feeding of *M. balthica* differed between the regions only when considering the cyanobacteria *N. spumigena*, where the southern clams feeding was higher, but not for the diatom *S. marinoi*. Diatoms are believed to be a more nutritious food source for benthic invertebrates, as they contain more essential fatty acids [60], and previous studies suggest a preference of diatoms over cyanobacteria by *M. balthica* [64]. Indeed, diatoms have been reported to increase growth rates of benthic crustaceans over cyanobacteria in the Baltic Sea [62]. Thus, the lack of difference in detected phytoplankton between the regions is not surprising, as all clams might recognize *S. marinoi* as an attractive food source. Additionally, multiple studies have showed that *M. balthica* [64,84] and other macrofauna [85] can incorporate organic matter from settling cyanobacteria to sediments. This incorporation occurs despite cyanobacteria's lower nutritional status as a food source for benthic and planktonic consumers [60,62]. While *N. spumigena* blooms have been occurring in the Baltic Sea since its formation [86], they are relatively patchy [87], and buoyant, resulting in a large part of the bloom being consumed in the water column [88]. [66] found that physical forcing exposure determined *M. balthica* feeding strategy; those in more wave-exposed stations relying more on suspension feeding, while those in wave-sheltered stations utilized deposit feeding, and that feeding strategy could be manipulated by transplanting the populations. Additionally, they found more and better quality organic matter, in the form of organic matter content, chlorophyll-*a* from phytoplankton and C:N ratios, was deposited in the sediment at the sheltered site, providing an incentive to deposit feed [66]. This would provide support for our findings of an adaptive response of southern *M. balthica* from the more sheltered station, which consumed more *N. spumigena*, as they likely spend more time deposit feeding. In this experiment, *N. spumigena* was mixed into the sediment in order to prevent floating from the cyanobacteria's gas vesicles, and thus were available to deposit feeding rather than suspension feeding, as would be present in the field. However, this result will require confirmation from additional studies; our clams were kept in lab conditions with no physical forcing for more than six months before the start of the experiment, so it is unlikely that physical forcing of their original station still determines their feeding strategy. Additionally, sediment samples from before, during, and after the experiment would be required to confirm the targeted phytoplankton DNA levels in the sediment. Indeed, the baseline of *N. spumigena* in the southern clams was higher than the northern clams before fresh phytoplankton were added, and these results must be taken with that consideration. Some

studies have previously shown viable diatoms remain in the sediment long after their sedimentation [89], and that bacterial degradation is required for some detritivores to consume diatoms [90]. However, these studies have relied on visual observation of gut and fecal content, which will favor the hard silica cell walls of diatoms over other types of phytoplankton [89,90], or longer-term integration of diet, such as stable isotopes or fatty acid analysis [91,92] which may not capture short-term bloom pulse events. Nevertheless, we were able to confirm previous findings that *M. balthica* does indeed consume both diatoms and cyanobacteria by newly developed molecular methods [64,84].

As far as we know this is the first study that employed both metabarcoding and qPCR to investigate the gut contents of benthic detritivores. Interestingly, *N. spumigena* and *S. marinoi* were also detected in all metabarcoding samples, but were not among the top phytoplankton in terms of relative abundance in the DNA metabarcoding dataset. In the metabarcoding dataset, we did not find significant differences in consumed phytoplankton communities by phytoplankton fed, but the two regional groups showed significantly different phytoplankton communities. This finding furthers our hypothesis that the *M. balthica* from different stations have different trophic strategies, but could also simply be an indication of different phytoplankton in the sediment, and would require analysis of the phytoplankton DNA present in the sediment before the additions of organic matter made in our experiment to make accurate conclusions. Indeed, the communities and relative abundances of phytoplankton ASVs found here were similar to communities found in oxic Baltic Sea sediments [93], indicating that *Synechococcus* is present in the sediment and ingested as a food source for benthic detritivores, potentially by both deposit and suspension feeding. Interestingly, our study indicates that phytoplankton DNA remains available in the sediment in ingestible form for at least six months after fresh organic matter addition, and is consumed by benthic detritivores. This serves to illustrate that much more work is required to understand diet of benthic detritivores.

In addition to the picocyanobacteria *Synechococcus*, we found large proportions of Trebouxiophyceae and Mamiellophyceae (green algae) in the guts of *M. balthica*. All three of these groups are unicellular phytoplankton, with little known about their ecological roles [94–96]. Dietary studies of *M. balthica* and other benthic detritivores in the Baltic Sea have focused on larger phytoplankton, such as chain-forming diatoms and filamentous cyanobacteria [64,84,85]. This previously unknown link from unicellular phytoplankton to an important bioturbator species indicates an uncovered benthic-pelagic link that might be important for food web stability and should be investigated further.

It is important to consider the mechanism of secondary predation here [97]; it is possible that some intermediate organism, such as meiofauna, is consuming the phytoplankton, and when the intermediate organism is consumed by *M. balthica* [52], we detect the phytoplankton that is not purposefully or directly consumed by *M. balthica*. This has been demonstrated in terrestrial invertebrates before, where secondary predation was detected for up to 8 hours after feeding (aphids fed to spiders, 4 hours later spiders fed to carabid beetles, and 4 hours later DNA from aphids was detected in the beetles; [35]). Additionally both phytoplankton species used in this study can release DNA, which is taken up by other organisms, such as bacteria. Then, when those bacteria are consumed by *M. balthica* in our experiment, we detected the *N. spumigena* DNA and interpret this result as *M. balthica* feeding on *N. spumigena*. Indeed, this has been proposed for meiofauna utilizing carbon from cyanobacteria [98], and even another species of *Macoma* genus in the Bering Sea [91]. In fact, [90] have suggested that diatoms require bacterial processing before being able to be used as an organic matter source. A potential solution for future methodological development is to focus on RNA of gut contents in addition to DNA. This has recently been shown to differentiate between recently ingested live prey and scavenged prey in beetles [99]. While secondary predation complicates the

interpretation of molecular trophic studies, one must consider the question of whether it matters; in the end, *M. balthica* derives energy from all organic sources consumed, whether on purpose or incidentally.

## Conclusions and future suggestions

Here we detected phytoplankton prey DNA in the guts of benthic detritivorous clams by means of qPCR and metabarcoding. Furthermore, we found that the signal of this feeding can be detected at least one week after ingestion by both qPCR and DNA metabarcoding. Our results suggest that the feeding of two regional groups of the Baltic clam on a diatom species do not differ, but clams from the more sheltered site fed significantly more on a cyanobacteria species, possibly indicating higher availability of cyanobacterial material or higher trophic plasticity that results in potential variations of feeding strategies. However, these fresh organic matter additions did not make up the majority of the clams' diet. Our experimental data suggests a significant proportion of the diet is made up of picocyanobacteria and unicellular green algae, which were not added freshly during the experiment. This indicates that phytoplankton DNA can remain in sediments for at least six months, and that these phytoplankton can sustain deposit-feeding *Macoma balthica*. Ecological models rely on detritivore diet and feeding rates determined by classical methods. Molecular tools could improve the resolution of these models and more accurately predict ecosystem functional changes with anthropogenic pressures, particularly as climate change is altering phytoplankton communities and input of organic matter to sediments.

## Supporting information

**S1 Fig. Map of stations utilized in the experiment.** Map of northern and southern Stockholm archipelago stations where *Macoma balthica* were collected to be utilized in the feeding experiment.
(PDF)

**S2 Fig. Relative abundance at the species level of samples.** Relative abundance of all species found with DNA metabarcoding of the 23S rRNA gene in *Macoma balthica* guts. Sample name colors indicate the feeding treatment, where gray indicates no feeding, green the diatom *Skeletonema marinoi* and brown the cyanobacteria *Nodularia spumigena*. The shape above the sample name indicates the region of *M. balthica*, where square represents the northern Stockholm archipelago clams and circle the southern Stockholm archipelago clams. The black bars above the relative abundance indicates the time point when the samples were taken in hours, with -24 h being before the feeding began, 0 h directly after feeding, and 24 h, 48 h, 72 h, and 144 h indicating 1, 2, 3, and 7 days after feeding, respectively.
(PDF)

**S3 Fig. Full phytoplankton community ordination.** NMDS using Bray-Curtis distance on DNA metabarcoding data from the guts of *Macoma balthica*, with filled shapes representing the southern Stockholm archipelago clams, and open shapes representing the northern Stockholm archipelago clams. Samples taken before the experiment began (Initial, circles) weren't fed, while squares represent feeding of the cyanobacteria *Nodularia spumigena* (Ns), and triangles the diatom *Skeletonema marinoi* (Sm) (stress = 0.16). Colors are the sampling time point in hours.
(PDF)

**S4 Fig. Phytoplankton in guts of *Macoma balthica* in cleared guts treatment.** Phytoplankton consumption by *M. balthica* region (northern A, B; southern C, D), with (dashed lines)

and without (solid or dotted lines) gut clearing, determined by qPCR and reported in number of copies $\mu L^{-1}$. Panels A and C represents *Skeletonema marinoi* (green), while B and D represents consumption of *Nodularia spumigena* (brown). Squares and solid lines represent the northern Stockholm archipelago clams, and circles and dotted lines represent the southern Stockholm archipelago clams. Error bars represent standard error.
(PDF)

**S1 Table. Sample information for qPCR analysis.** Sample is the *Macoma balthica* individual, region refers to the origin of the individual, phytoplankton fed refers to the phytoplankton culture fed to the individual, empty guts refers to the treatment where the individuals were placed in fresh water for 24 hours to empty their guts, time hours refers to the number of hours after feeding that the individuals were sampled, biological replicate refers to the individual replicate number, wet weight g refers to the wet weight of the individual in grams, CT AVG refers to the average $C_t$ value of the three technical replicates from the individual, CT SD refers to the standard deviation of the three technical replicates $C_t$ value from the individual, Log DNA refers to the log(10) amount of DNA detected from the qPCR assays, No copies refers to the number of copies of the target amplicon DNA, Copy no per µl refers to the number of copies of the target amplicon DNA standardized to the total DNA concentration, and sequenced refers to the selected samples which were also analyzed for next-generation sequencing to detect all phytoplankton in the guts, for more information on these samples see S2 Table.
(PDF)

**S2 Table. Sample information for the selected samples that underwent next generation sequencing.** Sample name is the *Macoma balthica* individual, biosample accession refers to the reference number to the sequencing data in the National Center for Biotechnology Information (NCBI) sequence read archive (SRA), region refers to phytoplankton fed refers to the origin of the individual, phytoplankton culture fed to the individual, empty guts refers to the treatment where the individuals were placed in fresh water for 24 hours to empty their guts, time hours refers to the number of hours after feeding that the individuals were sampled, and biological replicate refers to the individual replicate number.
(PDF)

**S1 File. Cleared guts experiment treatment results and discussion.** Within this experiment, we had an additional treatment to test the effectiveness of "gut clearing," the practice of allowing macrofauna purge their guts in fresh water for 24 hours. This is a common practice in stable isotope studies, and is intended to remove any recently digested items from the analysis to not disrupt the signal, as stable isotope analysis is primarily interested in the assimilated food items [9–12]. The statistical methods used are described in the main text, and presented and discussed here.
(PDF)

## Author Contributions

**Conceptualization:** Julie A. Garrison, Francisco J. A. Nascimento.

**Data curation:** Julie A. Garrison.

**Formal analysis:** Julie A. Garrison.

**Funding acquisition:** Francisco J. A. Nascimento.

**Investigation:** Julie A. Garrison.

**Methodology:** Julie A. Garrison, Nisha H. Motwani, Elias Broman.

**Project administration:** Julie A. Garrison.

**Resources:** Francisco J. A. Nascimento.

**Software:** Julie A. Garrison, Elias Broman.

**Supervision:** Francisco J. A. Nascimento.

**Visualization:** Julie A. Garrison.

**Writing – original draft:** Julie A. Garrison.

**Writing – review & editing:** Julie A. Garrison, Nisha H. Motwani, Elias Broman, Francisco J. A. Nascimento.

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
