## [Decision Letter · Decision Letter 0]

30 May 2022

PONE-D-22-11031Molecular diet analysis enables detection of diatom and cyanobacteria DNA in the gut of *Macoma balthica*PLOS ONE

Dear Dr. Garrison,

Thank you for submitting your manuscript to PLOS ONE. I have now received two reviews and I think they have provided you and your co-authors with some constructive suggestions that would improve the presentation and interpretation of the data presented. In particular, Reviewer 1 asks that you consider evidence that diatoms can remain viable in sediments for lengthy periods and require degradation by bacterial metabolism prior to ingestion by benthic invertebrates. This reviewer thinks that the study as presented is flawed, but suggests a workaround that would involve evaluating DNA preservation prior to the initiation of the experiment as a control to confirm the interpretation provided. Reviewer 2 has also provided a number of recommendations that will improve presentation of the study. I invite you to submit a revised version of the manuscript that addresses the points raised during the review process.

We look forward to receiving your revised manuscript.

Kind regards,

Lee W Cooper, Ph.D.

Section Editor

PLOS ONE

Journal Requirements:

Reviewers' comments:

Reviewer's Responses to Questions

**Comments to the Author**

1. Is the manuscript technically sound, and do the data support the conclusions?

Reviewer #1: No

Reviewer #2: Yes

2. Has the statistical analysis been performed appropriately and rigorously? 

Reviewer #1: No

Reviewer #2: Yes

3. Have the authors made all data underlying the findings in their manuscript fully available?

Reviewer #1: Yes

Reviewer #2: Yes

4. Is the manuscript presented in an intelligible fashion and written in standard English?

Reviewer #1: Yes

Reviewer #2: Yes

5. Review Comments to the Author

Reviewer #1: This paper is nicely written and informative, and addresses an important issue. Although the study was understandably exploratory, the results make it apparent that the experimental design was not suitable for indicating the retention time of phytoplankton DNA in clam guts. It would be good to see the information cast in a different light, perhaps about the response (or lack thereof) of DNA in deposit-feeder guts to inputs of fresh phytoplankton, given their apparent dependence on longer-term pools of organic matter (and associated DNA) in the sediments.

However, as the study is currently framed, it is critical to confirm that the “aged” sediments that the clams had been feeding on for the previous 6 mo did not contain persistent DNA from the same phytoplankton taxa that were later added to these sediments for the experiments. Viable chlorophyll can remain present in sediments for a number of months and even much longer (Pirtle-Levy et al. 2009. Deep-Sea Res II 56:1326). Moreover, at all times and especially during winter, a number of stable isotope, fatty acid, and mass-balance analyses have indicated that deposit-feeders such as Macoma must ingest appreciable amounts of bacteria that in turn have ingested older organic matter (mainly phytodetritus). Thus, “aged” sediments without fresh microalgal inputs for a number of months could still be a source of DNA from microalgae and cyanobacteria. This situation must in fact have been the case, because the experimental clams were maintained in the “aged” sediments for 6 mo before the feeding experiments began. Valid controls for the feeding experiments required sampling the clams’ guts before feeding began to see what they were already eating, and analyses of the sediments from before and throughout the experiments to infer any selectivity for cyanobacteria vs. diatoms. The fact that no temporal trends were detected in DNA from fed phytoplankton in the guts over time, and that the fed taxa comprised a relatively moderate or even small fraction of the taxa consumed, suggests that the clams simply continued to feed as they had before with little effect of the added materials.

Moreover, the authors attribute greater cyanobacterial consumption by Macoma to higher cyanobacterial production in the more hydrographically sheltered, shallower sampling site. However, samples of clams and sediments at the shallower site were collected in October, whereas samples at the deeper site were collected 3 mo later in January. The pool of fresh microalgae and cyanobacteria in the sediments can change a lot over 3 mo, although the DNA present in the longer-term organic matter pool may not. The relative availability of DNA from cyanobacteria and eukaryotic algae of these two samples might reflect seasonal differences between sampling periods rather than site-specific effects. Again, DNA analyses of the sediments themselves would have clarified interpretation of variations in the animals’ guts. Perhaps I misunderstand your analysis, but if the fed taxa were already present in the clams’ diets, feeding experiments will say nothing about retention of the DNA signal (presence-absence), although they might indicate a response in total consumption.

Perhaps the authors have data on the relative availability of DNA of different phytoplankton taxa in the sediments and from the guts of clams before the experiments began. If you have such data, this experiment may in fact say more about how long it will take relative food intake (amounts of DNA) to respond to food addition, rather than how long the response will last. The authors must at least recognize these issues and convince the reader that these caveats are not important shortcomings of their analysis.

1. L 29. Briefly explain what "aged sediment" is. How long had the sediments been removed from the field? Live chlorophyll can often be detected for a number of months and perhaps years after being deposited in sediments (see above general comments). This issue is critical to the validity of your experiment, so more explanation is needed. Did you test for DNA of phytoplankton in this "aged" sediment?

2. L 96. “Detritus” is defined differently by various authors, and in recent food web models is usually designated as a separate compartment from living bacteria. If you mean the term “detritus” to include bacteria, please say so in parentheses.

3. L 119. Replace "feeding rates of" by "rates of feeding on"

4. L 128-129. Awkward wording with unclear meaning

5. L 135. The Introduction is nicely written and informative. I suggest, however, that you include mention of the many papers on diets of Macoma and other deposit-feeders, and especially the inference that that they also ingest cells and exudates of bacteria that subsist on a longer-term pool of organic matter derived from phytodetritus (which contains its DNA) (see Deep-Sea Res II 102:84 (2014), Ecol Applic 24:1525 (2014), and references therein).

6. L 152-153, 162-163. If the clams were kept for 6 mo without being fed, they must have been eating organic matter (phytodetritus, bacteria) in the sediments during this period. Doesn’t this mean that in fact there were persistent remnants of phytoplankton and their DNA in the sediments? Moreover, the fact that the del13C of bulk sediment organic matter is often much higher than that of settling particulate matter, while the del13C in deposit-feeder tissues is more similar to that of sediments, suggests that most phytodetritus is substantially reworked by bacteria before being assimilated by deposit-feeders (see above references). Thus, remnant DNA of phytoplankton in “aged” sediments could be quite important in the guts of deposit-feeders. Some experiments have shown that despite consumption of fresh diatoms, the fresh diatoms are often not initially assimilated and must be conditioned by bacteria or by multiple passage through the guts of invertebrates before assimilation (J. Exp Mar Biol Ecol 308:59, 2004). The phytoplankton used in your feeding experiments were in fact not among the top taxa detected (L 323-327, 355-356). This result seems to indicate that there was much residual DNA from other species in the “aged” sediments that was consumed by Macoma, so that it’s hard to judge selectivity or the duration of detection of fed phytoplankton in the animals without knowing availability in the sediments. You summarize the latter issue well on L 355-359, which in effect says that your experiment provided little information on longevity of detection, which appears to be much longer than your experiments.

7. L 188-189. Given that published studies of some animals had shown persistence of DNA in the guts for up to 5 days and perhaps longer than 6 days (366-368), it was certainly possible that 24 hours was not long enough for Macoma to empty its gut of DNA from food. The standard 24-h wait to allow emptying of the gut is for whole-body stable isotope analyses that are not so sensitive to very small amounts of material. As your study revealed, your 24-h “negative controls” were not reliable controls, an important finding but one which eliminated their usefulness as controls in this experiment. A better control would be to analyze the gut contents before experimental feeding, when the clams had been subsisting on longer-term phytodetritus-derived DNA in the sediments which included the phytoplankton taxa you fed them. That way you could compare the DNA in the guts that would have been there anyway vs. any changes owing to feeding. It could be that the diatoms you were feeding the clams were not being digested and that the clams were simply continuing to feed on the diatom-derived organic matter that had sustained them for the preceding 6 mo before feeding. That might be why you saw no temporal trends in DNA from the fed phytoplankton. Were your Day Zero samples taken a number of hours after feeding, or rather just before you fed the clams?

8. L 243. I don’t think you ever define the acronym ASV.

9. L 313-314. Samples of clams and sediments from the southern population were collected in October vs. January for the northern population. This seasonal difference may confound the comparisons by depth or by degree of hydrographic sheltering. Data on availability of DNA of the two phytoplankton types in sediments before the experiments began would avoid this issue.

10. Figure 2. The fonts in this figure are far too small to be read easily.

11. L 397-399. See Comment 9 above. Also, the meaning of “mixed into the sediment in order to prevent floating” implies that cyanobacteria have evolved a special ability to mix into sediments that has not been evolved in diatoms, rather than such mixing being mostly a function of bioturbation or other mixing factors that could affect cyanobacteria and diatoms similarly. This novel concept needs confirmation by a citation.

12. L 411-412. Without knowing the relative abundance of cyanobacterial and diatom DNA in the sediments, you cannot say that this difference between clams from the two stations did not result simply from differences in availability in sediments at the two sites, perhaps due to the 3-mo difference in sampling time. The experimental clams had been surviving on available organic materials in the sediments for 6 months without additions of fresh phytoplankton. In some experiments and field biomarker studies (see above), it appeared that fresh phytoplankton cells were consumed opportunistically and perhaps somewhat incidentally, and were not immediately important to assimilation; thus, fresh cells of particular taxa might not be an actual target of feeding. One cannot conclude that one population relied more on filter-feeding based on lower ingestion of cyanobacteria, given that both foods can be ingested by either filter-feeding or deposit-feeding and the samples were taken 3 mo apart.

13. L 424-426. Single-celled green algae have been reported in Macoma gut contents previously, but did not appear abundant enough to be considered an important food (see above references). However, your DNA analyses provide new insights into possibly greater importance of these algae.

14. L 445-446. High trophic plasticity, but not necessarily different feeding strategies. As noted above, you must confirm that availability of cyanobacterial DNA in the sediments did not differ between seasons of sampling to conclude selectivity by the southern population sampled in autumn vs. the northern population sampled in midwinter.

Reviewer #2: Overview

The manuscript looks at using molecular techniques to examine diet of a common bivalve that has multiple feeding strategies and thus can potentially sit at many places trophically. Using molecular techniques can help overcome some of the pitfalls found in isotope analyses more traditionally used for this work. Overall this paper provides a lot of new information on a group of organisms where information can be lacking, benthic invertebrates. I think this manuscript can add to the scientific literature, once some of the below comments are addressed.

Major Comments

Note that the references as cited in text and in the reference section don’t match the PLOS One style of citation. Please see https://journals.plos.org/plosone/s/submission-guidelines#loc-references

“References are listed at the end of the manuscript and numbered in the order that they appear in the text. In the text, cite the reference number in square brackets (e.g., “We used the techniques developed by our colleagues [19] to analyze the data”). PLOS uses the numbered citation (citation-sequence) method and first six authors, et al.”

Adding a figure of a map where station locations are shown would be helpful in addition to the listing of the latitudes and longitudes. It will illustrate the difference between the sheltered and open sites better as well and where they are located in space.

Why did you pool stations for collection? Why did the northern cluster have more stations than the southern cluster? 24 km between the northern stations seems like it could make a difference in physical characteristics like the phytoplankton blooms in the region or the physical forcings you discuss? Is that possible? Please elaborate more on why you chose to pool the stations together.

Where were the samples held/experiments conducted after collection? An incubator at the university? Or at a different location.

Line 176-177 How did you arrive at those concentrations of each of the phytoplankton species to use? I would like to see the same justification for those values as you use for the number of clams you put into each mesocosm (which was great). Additionally, is there a more recent estimate of the number of individuals per meter squared in addition to the 1976 reference? Populations can change pretty drastically over time, so are the values the same ~40 years later?

Why are the mesocosms and sediment heights different sizes in the fed vs unfed (lines 178 and 182)?

Line 151-153 How often was water replenished? Was this done manually or through a flow through system? What did maintaining this for the 6 months before the experiments started look like? i.e. daily checks on salinity? Temperature? Water changes? Etc.

In the discussion, particularly the section “Feeding of the Baltic clam on two phytoplankton species”, I would like to see a little more on why you think cyanobacteria was so present when it is the less nutritious food. I think lines 386-390 is trying to do that, but it is a little confusing to follow. – see note below in minor comments about that particular sentence.

Please be consistent when spelling out the genus of the organisms vs. using the abbreviated start – see specific line comments below under minor comments.

The figures all appear to be a little bit blurry, please try and fix the resolution.

I would consider adding your results, discussion, and figure on the gut clearing analysis in the main text. I think the arguments you make there help to justify why this method in addition to stable isotope analyses is important, a point you bring up in the introduction of the manuscript.

Minor Comments

Key words – Limecola balthica vs. Macoma balthica in the title. May be a bit confusing, but since they are the same species, perhaps it is to maximize the ability for people to find the paper? Could the key word look like it did in the text Macoma (Limecola) balthica?

Line 26, 27, 28 – Can you include the taxonomic citation for the species? For example Macoma calcarea (Gmelin, 1791)

Line 29 – What is meant by aged sediment? Can you be more specific please?

Line 65 Add “such as” before “soft”

Line 68 Change “references” to “reference”

Line 65/66 Adding a few more details about why they are not readily observable, for example is it because the soft bodied prey decay faster?

Line 80 Remove the extra space after the “(“

Line 94 Which greenhouse gases?

Line 94, 95, 108 When you start the sentence with the genus, spell out the full genus – please replace “M.” with Macoma” – there are a few other instances throughout the manuscript of this, please fix and standardize all.

Line 95 Please list examples of the fish species that eat M. balthica

Line 98 Can you provide a definition for spring in the Baltic, what months does the bloom generally settle? Same when you mention the summer bloom. Do these blooms overlap at all? Or in space and time are they distinct from one another?

Line 101 and 103 Are there values to more specifically describe how much one bloom is increasing in magnitude vs how much the other is decreasing? If yes, could you please add to quantify what increasing and decreasing mean in this ecosystem?

Line 120 Remove the word “which”

Line 123 Macoma can be abbreviated to M.

Line 125 Is the Hedberg et al. 2020 citation in reference to just the diatom prey or both the cyanobacteria and the diatom? If it is not for both, please include a reference for cyanobacteria as prey for these clams

Line 126 and 127 You can abbreviate the genus name of the phytoplankton species and the clam after using the genus name fully one as you do in line 125 – when it is not the beginning of the sentence

Line 116 and 128 Can you please provide some more details about what you mean when you use the term “trophic plasticity”? Is it that they can have flexibility in where they are feeding because of the different feeding strategies and the broad array of prey?

Line 135 Can you please elaborate on a few days and provide a more specific window, for example 2-4 days or 3-5 days?

Lines 138-142 and Lines 144-147 These sentences are a little long and clunky to read, please revise, perhaps into more than one sentence

Line 142 Replace “see Fig 1 for timeline” with “Fig 1”

Line 144 Spell out Macoma for the start of the sentence

Line 156 Southern does not need to be capitalized

Line 239 Please also include the version of R you were using (like you do in line 255) in addition to the version of the package

Line 243 Please define ASV at its first use in the manuscript

Line 283 What do you mean difference in DNA- the amount was the same at every time point?

Line 283 Add the word “significant” between “no” and “differences”

Line 299 In all of the table and figure captions spell out genus names – and keep it consistent, here you spell out the phytoplankton genus names but not Macoma. If you mention the name a second time within the caption you can use the abbreviated start for the genus.

Line 313 – You write, significantly higher, please list what the p value was in parentheses behind amounts or point to the specific part of Table 1 you are referencing, it is a little confusing

Line 347 What is the Stress=0.16? It isn’t a complete sentence so should be added to another sentence or more explanation should be added to make it a complete sentence.

Line 386 Define OM the first time you use it

Line 386-391 This sentence is hard to follow please revise

Line 391 This should be the start of a new paragraph – if it is then indenting paragraphs throughout the manuscript would make this clearer

Line 394 Organic matter is spelled out here where earlier you used OM, please be consistent, either spell it out every time or define it and then use OM throughout the rest of the discussion

Line 408-411 What is meant here by phytoplankton fed? I think I understand this sentence, but it gets a little confusing, can you add some details and specifics please.

Line 411 After communities add “of phytoplankton”

Line 428 What kind of intermediate organisms might M. balthica be consuming, can you provide some examples? And are those animals feeding on the species you found inside the clams?

Supplementary Material

Line 4 Add “to” between “macrofauna” and “purge”

6. PLOS authors have the option to publish the peer review history of their article (what does this mean?). If published, this will include your full peer review and any attached files.

Reviewer #1: No

Reviewer #2: No

---

## [Author Response · Author response to Decision Letter 0]

31 Aug 2022

Thank you for submitting your manuscript to PLOS ONE. I have now received two reviews and I think they have provided you and your co-authors with some constructive suggestions that would improve the presentation and interpretation of the data presented. In particular, Reviewer 1 asks that you consider evidence that diatoms can remain viable in sediments for lengthy periods and require degradation by bacterial metabolism prior to ingestion by benthic invertebrates. This reviewer thinks that the study as presented is flawed, but suggests a workaround that would involve evaluating DNA preservation prior to the initiation of the experiment as a control to confirm the interpretation provided. Reviewer 2 has also provided a number of recommendations that will improve presentation of the study. I invite you to submit a revised version of the manuscript that addresses the points raised during the review process.

Dear Dr. Cooper,

Thank you for the opportunity to revise the manuscript “Molecular diet analysis enables detection of diatom and cyanobacteria DNA in the gut of Macoma balthica.” We appreciate the work that the reviewers have put into our manuscript and have taken the comments seriously and addressed each comment below. 

Briefly, we agree with Reviewer 1 that sediment samples before the start of the experiment would have been beneficial to determine which food sources were present and available to the clams, but we do provide samples of the clam gut contents before the feeding part of experiment (-24 h samples). With regard to the reviewer’s concern that the populations differed in terms of available diatoms from the start, the populations are not significantly different before feeding in the diatom qPCR assay (Fig 2A). Thus, we do not believe that there were inherent differences between the sediments in terms of diatoms, but other phytoplankton groups likely differed from the beginning. We recognize this limitation and have added this to the discussion in lines 396-403. We have also provided a discussion on the phenomenon of diatoms requiring bacterial degradation prior to consumption by M. balthica (secondary predation, lines 426-443) and provided a suggested future workaround to determine fresh vs. dead ingested organic material in the form of RNA vs. DNA. This technique has recently been shown to work in predatory and scavenger beetles (Neidel et al. 2022).

We have also implemented most of Reviewer 2’s suggestions, and provided responses to those below. Briefly, we have updated several inconsistencies that Reviewer 2 pointed out, including updating the manuscript to the correct PLOS ONE citation, updating several references, and streamlining our species nomenclature. We are appreciative of their detailed comments which have improved the readability of the manuscript.

We believe that the comments by the reviewers have improved the manuscript, and better explained the benefits and challenges of these new molecular methods. We have provided more nuance to our discussion about the kinds of questions that DNA, qPCR and metabarcoding can and cannot answer, and potential for complimentary food web methods. Please note that line numbers refer to the manuscript with changes tracked in order to facilitate easier understanding of changes made.

Sincerely,

Julie Garrison and co-authors

 

Reviewer #1

This paper is nicely written and informative, and addresses an important issue. Although the study was understandably exploratory, the results make it apparent that the experimental design was not suitable for indicating the retention time of phytoplankton DNA in clam guts. It would be good to see the information cast in a different light, perhaps about the response (or lack thereof) of DNA in deposit-feeder guts to inputs of fresh phytoplankton, given their apparent dependence on longer-term pools of organic matter (and associated DNA) in the sediments.

Thank you for reviewing our manuscript. We agree that the experiment was ultimately too short to adequately address the aim of determining the retention time of phytoplankton DNA in the guts of Macoma balthica, but this novel finding contradicts previous estimates for gut retention time, and thus we believe it adds valuable information warrant publication that following studies can build upon in the future. As this was the first study investigating M. balthica diet by molecular methods that we are aware of, we were unsure how long of an experiment would be required to detect the retention time, and based our experimental length off of previous studies on similar organisms. We appreciate your suggestions for redirected focus of the aims and discussion, and we have reduced the focus on the aim of retention time determination through removing this aim from primary to secondary aim in lines 114-118. Additionally, we have placed more focus on confirming the presence of phytoplankton DNA in the guts in lines 111-114. For your convenience, the cited line numbers refer to the tracked changes copy of the manuscript.

However, as the study is currently framed, it is critical to confirm that the “aged” sediments that the clams had been feeding on for the previous 6 mo did not contain persistent DNA from the same phytoplankton taxa that were later added to these sediments for the experiments. Viable chlorophyll can remain present in sediments for a number of months and even much longer (Pirtle-Levy et al. 2009. Deep-Sea Res II 56:1326). Moreover, at all times and especially during winter, a number of stable isotope, fatty acid, and mass-balance analyses have indicated that deposit-feeders such as Macoma must ingest appreciable amounts of bacteria that in turn have ingested older organic matter (mainly phytodetritus). Thus, “aged” sediments without fresh microalgal inputs for a number of months could still be a source of DNA from microalgae and cyanobacteria. This situation must in fact have been the case, because the experimental clams were maintained in the “aged” sediments for 6 mo before the feeding experiments began. Valid controls for the feeding experiments required sampling the clams’ guts before feeding began to see what they were already eating, and analyses of the sediments from before and throughout the experiments to infer any selectivity for cyanobacteria vs. diatoms. The fact that no temporal trends were detected in DNA from fed phytoplankton in the guts over time, and that the fed taxa comprised a relatively moderate or even small fraction of the taxa consumed, suggests that the clams simply continued to feed as they had before with little effect of the added materials.

We agree with the reviewer that in the field, chlorophyll a can remain available in the deeper sediments for extended periods of time. However, we conducted our experiment in a laboratory setting, and the clams had a maximum sediment depth of 4 cm available to them, which M. balthica has been shown to burrow beyond this depth before (6 cm; Edelaar et al. 2003). Thus, M. balthica was able to feed at all sediment depths present and it is less likely that diatoms would be able to remain uneaten in the experiment sediment compared to field conditions. Additionally, there was no sedimentation of fresh organic matter during the 6-9 months before the start of the experiment. As hypothesized by Pirtle-Levy et al. 2009, the deeper organic matter reserves are available for the periods during which there is no input of fresh organic matter, such as winter to early spring, when our samples were isolated in the lab. Additionally, the phytoplankton utilized in this study sediment with the spring (April, diatoms) and summer (August-September, cyanobacteria), meaning that the DNA from the selected phytoplankton would need to be present in the sediment for about a year from sedimentation in the field to incubation to experiment. We find this unlikely, especially from the data that our clams do indeed consume these phytoplankton species, and have been shown to immediately react to and incorporate fresh organic material (Ólafsson et al. 1999; Karlson et al. 2010; Nascimento et al. 2012; Hedberg et al. 2020). Additionally, any DNA remaining in the sediments when the experiment began should be very small compared to the fresh phytoplankton that we added. We have added the possibility of DNA remaining from the field to our discussion in lines 396-403, which reads “Additionally, sediment samples from before, during, and after the experiment would be required to confirm the targeted phytoplankton DNA levels in the sediment. Some studies have previously shown viable diatoms remain in the sediment long after their sedimentation [89], and that bacterial degradation is required for some detritivores to consume diatoms [90]. However, these studies have relied on visual observation of gut and fecal content, which will favor the hard silica cell walls of diatoms over other types of phytoplankton [89,90], or longer-term integration of diet, such as stable isotopes or fatty acid analysis [91,92] which may not capture short-term bloom pulse events.”

We have clam samples from before the feeding occurred (-24 hour samples) and present them in our paper. Unfortunately, our samples of the sediment were not stored properly and thus not viable for DNA analysis, and we agree that they would have contributed greatly to this experiment.

In order to have adequate controls for such an experiment, we would need to present the clams an unnatural environment, such as sterilized sediment, completely empty any existing microbiota from the clams’ digestive tract, and constantly move the clams to new sterilized sediment as to prevent the consumption of their own feces or external microbiota. This was considered, but we were concerned that M. balthica individuals would be stressed in such sediments, and would decrease or completely arrest its feeding during our experimental period. This experiment potentially would answer a very specific question about food DNA passing through the digestive tract of the clams, but would not be applicable to the natural environment.

Moreover, the authors attribute greater cyanobacterial consumption by Macoma to higher cyanobacterial production in the more hydrographically sheltered, shallower sampling site. However, samples of clams and sediments at the shallower site were collected in October, whereas samples at the deeper site were collected 3 mo later in January. The pool of fresh microalgae and cyanobacteria in the sediments can change a lot over 3 mo, although the DNA present in the longer-term organic matter pool may not. The relative availability of DNA from cyanobacteria and eukaryotic algae of these two samples might reflect seasonal differences between sampling periods rather than site-specific effects. Again, DNA analyses of the sediments themselves would have clarified interpretation of variations in the animals’ guts. Perhaps I misunderstand your analysis, but if the fed taxa were already present in the clams’ diets, feeding experiments will say nothing about retention of the DNA signal (presence-absence), although they might indicate a response in total consumption.

Both areas were below the photic zone and sediments were collected during the low sedimentation period of fresh organic matter for the Baltic Sea (see below figure of pelagic phytoplankton carbon throughout a typical year close to our sampling sites from the Swedish national monitoring program sharkweb.smhi.se; both October and January are outside of the main phytoplankton peaks, with October having slightly higher phytoplankton biomass than January, but was also incubated without added food for 3 months longer). As mentioned previously, our expectation based on available literature was that the added fresh phytodetritus would 1) be quantitatively larger than the small amounts of diatoms and cyanobacteria potentially left in the sediment, and 2) preferentially selected and fed on by M. balthica (Ólafsson et al. 1999; Nascimento et al. 2012; Hedberg et al. 2020).

Perhaps the authors have data on the relative availability of DNA of different phytoplankton taxa in the sediments and from the guts of clams before the experiments began. If you have such data, this experiment may in fact say more about how long it will take relative food intake (amounts of DNA) to respond to food addition, rather than how long the response will last. The authors must at least recognize these issues and convince the reader that these caveats are not important shortcomings of their analysis.

Unfortunately, we do not possess data on the sediment, but we do present the data from the guts of the clams as requested by the reviewer. We have tried to address the concerns of the reviewer by adding a section on the discussion with the caveats requested. It now reads “Additionally, sediment samples from before, during, and after the experiment would be required to confirm the targeted phytoplankton DNA levels in the sediment.” in lines 396-398. 

1. L 29. Briefly explain what "aged sediment" is. How long had the sediments been removed from the field? Live chlorophyll can often be detected for a number of months and perhaps years after being deposited in sediments (see above general comments). This issue is critical to the validity of your experiment, so more explanation is needed. Did you test for DNA of phytoplankton in this "aged" sediment?

The phrasing has been modified to “transferred back into their original sediment” instead, in order to explain how long the sediment had been removed from the field (6 or 9 months, the same as the clams). Unfortunately, the samples of the sediment were not viable as explained before, we agree that they would have contributed greatly to our results.

2. L 96. “Detritus” is defined differently by various authors, and in recent food web models is usually designated as a separate compartment from living bacteria. If you mean the term “detritus” to include bacteria, please say so in parentheses.

We agree that bacteria are not inherently included in the term detritus, and have thus made the suggested change.

3. L 119. Replace "feeding rates of" by "rates of feeding on"

The suggested change has been made.

4. L 128-129. Awkward wording with unclear meaning

The second aim has been revised for clarity, and now reads “ii) if feeding by the two M. balthica populations on each of the phytoplankton species differs, which would likely be due to trophic plasticity and physical forcing influences on their feeding strategy” in lines 116-118.

5. L 135. The Introduction is nicely written and informative. I suggest, however, that you include mention of the many papers on diets of Macoma and other deposit-feeders, and especially the inference that that they also ingest cells and exudates of bacteria that subsist on a longer-term pool of organic matter derived from phytodetritus (which contains its DNA) (see Deep-Sea Res II 102:84 (2014), Ecol Applic 24:1525 (2014), and references therein).

Thank you, we have reviewed the suggested articles and added relevant references. We have also included bacteria to the list of food items of M. balthica (lines 83-85), and already cite many studies of M. balthica diet determined by microscopy of stomach contents, feeding experiments, stable isotopes, and fatty acids. Secondary predation is the phenomenon that you bring up here, where M. balthica may not be directly consuming the phytoplankton DNA that we detect in their guts, but rather an intermediate organism (such as bacteria) have ingested it and we are detecting a feeding chain rather than direct link. This is a common problem not only for the techniques here utilized but also for other methods commonly used in ecology, such as radiotracers or stable isotopes. We agree with the reviewer that it is important to discuss this limitation also in the case of molecular techniques, and work towards a technical solution in the future. This problem is discussed within the section of the discussion regarding secondary predation (lines 426-443). 

This is a very relevant point and we thank the reviewer for the interesting references. However, we feel that these results and technique fit better in the discussion section. Therefore, we have added this suggestion to our final section containing suggestions for future studies (lines 438-440), including a reference to a recent study showing promising results from an even newer technique, utilizing RNA from the guts of scavengers to determine the most recently consumed prey (Neidel et al. 2022). We are optimistic that this technique could help determine if detritivores are consuming recently deposited phytoplankton or phytoplankton that are part of the old, reworked organic matter stores.

6. L 152-153, 162-163. If the clams were kept for 6 mo without being fed, they must have been eating organic matter (phytodetritus, bacteria) in the sediments during this period. Doesn’t this mean that in fact there were persistent remnants of phytoplankton and their DNA in the sediments? 

M. balthica was likely consuming the existing organic matter from the sediment during their 6-9 month incubation period. There was likely remnants of phytoplankton and their DNA in the sediments, but since there was no phytoplankton addition during this time, it is likely that the refractory organic matter was consumed, and that attractive food sources such as fresh phytoplankton were unavailable or very scarce, and we should therefore mostly be detecting the phytoplankton DNA of the freshly added material.

Moreover, the fact that the del13C of bulk sediment organic matter is often much higher than that of settling particulate matter, while the del13C in deposit-feeder tissues is more similar to that of sediments, suggests that most phytodetritus is substantially reworked by bacteria before being assimilated by deposit-feeders (see above references). Thus, remnant DNA of phytoplankton in “aged” sediments could be quite important in the guts of deposit-feeders. Some experiments have shown that despite consumption of fresh diatoms, the fresh diatoms are often not initially assimilated and must be conditioned by bacteria or by multiple passage through the guts of invertebrates before assimilation (J. Exp Mar Biol Ecol 308:59, 2004). The phytoplankton used in your feeding experiments were in fact not among the top taxa detected (L 323-327, 355-356). This result seems to indicate that there was much residual DNA from other species in the “aged” sediments that was consumed by Macoma, so that it’s hard to judge selectivity or the duration of detection of fed phytoplankton in the animals without knowing availability in the sediments. You summarize the latter issue well on L 355-359, which in effect says that your experiment provided little information on longevity of detection, which appears to be much longer than your experiments.

We recognize the importance of bacterial degradation for detritivore subsistence, and have clarified their importance and added the Hansen and Josefson 2004 reference within our discussion in lines 436-438, reading “In fact, [90] have suggested that diatoms require bacterial processing before being able to be used as an organic matter source.” Regarding the possible presence of phytoplankton and their DNA in the sediments before the start of the experiment, 6-9 months would have been ample time for bacterial reworking and several gut passages of the phytoplankton DNA, removing the possibility of detecting “old” DNA, and thus we should be detecting mostly “fresh” DNA of the added phytoplankton. However, as we discovered in the DNA sequencing, DNA from phytoplankton that were not added fresh to our experiment dominated the DNA in the guts of M. balthica. As we have not added this DNA, it must be coming from the existing sediment, and thus we recommend further investigations on whether this is “normal” preferred diet of M. balthica, or an enclosure effect of the experiment. We agree that we cannot determine the longevity of phytoplankton detection from our experiment, and have removed the focus from this aim. However, our study does provide new information regarding the diet of M. balthica, with the chosen phytoplankton to be fed to M. balthica for this experiment have been shown to constitute a significant proportion of their diet from previous studies, based on microscopy of stomach contents, feeding experiments, and stable isotopes (Ólafsson et al. 1993; Hedberg et al. 2020). Molecular methods such as those utilized here uncover new links, but will require some additional studies to fully understand.

7. L 188-189. Given that published studies of some animals had shown persistence of DNA in the guts for up to 5 days and perhaps longer than 6 days (366-368), it was certainly possible that 24 hours was not long enough for Macoma to empty its gut of DNA from food. The standard 24-h wait to allow emptying of the gut is for whole-body stable isotope analyses that are not so sensitive to very small amounts of material. As your study revealed, your 24-h “negative controls” were not reliable controls, an important finding but one which eliminated their usefulness as controls in this experiment. A better control would be to analyze the gut contents before experimental feeding, when the clams had been subsisting on longer-term phytodetritus-derived DNA in the sediments which included the phytoplankton taxa you fed them. That way you could compare the DNA in the guts that would have been there anyway vs. any changes owing to feeding. It could be that the diatoms you were feeding the clams were not being digested and that the clams were simply continuing to feed on the diatom-derived organic matter that had sustained them for the preceding 6 mo before feeding. That might be why you saw no temporal trends in DNA from the fed phytoplankton. Were your Day Zero samples taken a number of hours after feeding, or rather just before you fed the clams?

We agree with your statement that DNA has been shown to remain in the guts for longer than 5 days in other animal models, and we did not expect that 24 hour after feeding samples would be negative controls. Instead, we actually use controls exactly as the reviewer suggested: individuals sampled before feeding, termed -24 hour samples, to be used as baseline controls for fresh phytoplankton feeding material, as you have suggested. We found phytoplankton DNA even in these -24 hour controls, indicating that the clams were consuming some phytoplankton DNA from their original sediment, which had been in the sediment when sampled in the field. The clams were in the presence of added fresh phytoplankton material for 24 hours, from -24 to 0 hours. The -24 hour samples were taken just before feeding, and the 0 hour samples were taken directly after removing the clams to the no-fresh phytoplankton addition sediment. Figure 1 provides a timeline which presents the samples in relation to feeding, and we have clarified when the samples were taken in relation to feeding in the text in lines 177-180.

8. L 243. I don’t think you ever define the acronym ASV.

The definition “amplicon sequence variants” has now been added to line 240. 

9. L 313-314. Samples of clams and sediments from the southern population were collected in October vs. January for the northern population. This seasonal difference may confound the comparisons by depth or by degree of hydrographic sheltering. Data on availability of DNA of the two phytoplankton types in sediments before the experiments began would avoid this issue.

Yes, we agree that the difference in sampling times could present differences in the DNA detected in the clam guts, and that sediment DNA samples would have solved this issue. Please see our replies above to comments 1, 5, and 6. As mentioned we included samples of clams from before feeding. Additionally, late autumn and winter are periods of low primary productivity in the Baltic Sea as shown above, meaning that there was no significant settling of phytoplankton between the sampling occasions. The six month minimum incubation should have been adequate for the clams to consume the fresh organic matter in the sediments.

10. Figure 2. The fonts in this figure are far too small to be read easily.

The font size has been increased on this figure.

11. L 397-399. See Comment 9 above. Also, the meaning of “mixed into the sediment in order to prevent floating” implies that cyanobacteria have evolved a special ability to mix into sediments that has not been evolved in diatoms, rather than such mixing being mostly a function of bioturbation or other mixing factors that could affect cyanobacteria and diatoms similarly. This novel concept needs confirmation by a citation.

The comment “mixed into the sediment in order to prevent floating” refers to the experimental setup, where the phytoplankton cultures were mixed with sediment. Many cyanobacteria have well-known gas vesicles, which aid in buoyancy regulation. If the cyanobacteria were not mixed with sediment, they would float on the surface of the mesocosm water column and would not be available as food for the clams. This idea has been revised, and now reads “This would provide support for our findings of an adaptive response of southern population of M. balthica from the more sheltered station which consumed more N. spumigena, as they likely spend more time deposit feeding. In this experiment, N. spumigena was mixed into the sediment in order to prevent floating from the cyanobacteria’s gas vesicles, and thus were available to deposit feeding in lines 391-393. We hope that this revision clarifies any confusion.

12. L 411-412. Without knowing the relative abundance of cyanobacterial and diatom DNA in the sediments, you cannot say that this difference between clams from the two stations did not result simply from differences in availability in sediments at the two sites, perhaps due to the 3-mo difference in sampling time. The experimental clams had been surviving on available organic materials in the sediments for 6 months without additions of fresh phytoplankton. In some experiments and field biomarker studies (see above), it appeared that fresh phytoplankton cells were consumed opportunistically and perhaps somewhat incidentally, and were not immediately important to assimilation; thus, fresh cells of particular taxa might not be an actual target of feeding. One cannot conclude that one population relied more on filter-feeding based on lower ingestion of cyanobacteria, given that both foods can be ingested by either filter-feeding or deposit-feeding and the samples were taken 3 mo apart.

We acknowledge your statement about not knowing the initial availability of phytoplankton in the sediments, and have added this information to our discussion, which reads “Additionally, sediment samples from before, during, and after the experiment would be required to confirm the targeted phytoplankton DNA levels in the sediment” (lines 396-398). Indeed, as the DNA metabarcoding data showed significant community differences by population, and the largest components of the gut communities were not the fed phytoplankton cultures, it is likely that the sediments differed by food sources available. However, we do not see a significant difference by population before feeding (Line 328; PERMANOVA; F1,19 = 0.84, p = 0.64; S2 figure, -24 hour samples). The significant difference between populations by DNA metabarcoding communities emerges only after feeding.

13. L 424-426. Single-celled green algae have been reported in Macoma gut contents previously, but did not appear abundant enough to be considered an important food (see above references). However, your DNA analyses provide new insights into possibly greater importance of these algae.

While these food sources have been reported for other species in the genus Macoma in the cited references, this is the first time that they have been reported for M. balthica. We agree that our DNA results can provide more complete understanding of M. balthica diet and has implications for Macoma energetics, which we highlight in the discussion, for example in lines 423-425 and 453-456.

14. L 445-446. High trophic plasticity, but not necessarily different feeding strategies. As noted above, you must confirm that availability of cyanobacterial DNA in the sediments did not differ between seasons of sampling to conclude selectivity by the southern population sampled in autumn vs. the northern population sampled in midwinter.

We agree that differences in measured DNA concentrations in our experiment do not necessarily indicate differences in high trophic plasticity or feeding strategies, and we have edited this sentence to now read “Our results suggest that the feeding rates of two populations of the Baltic clam on a diatom species do not differ, but the population from the more sheltered site fed significantly more on a cyanobacteria species, possibly indicating high trophic plasticity from potentially variations of feeding strategies.” We have added the limitation of lacking sediment samples to several parts of the article, as indicated previously.

 

Reviewer #2 

The manuscript looks at using molecular techniques to examine diet of a common bivalve that has multiple feeding strategies and thus can potentially sit at many places trophically. Using molecular techniques can help overcome some of the pitfalls found in isotope analyses more traditionally used for this work. Overall this paper provides a lot of new information on a group of organisms where information can be lacking, benthic invertebrates. I think this manuscript can add to the scientific literature, once some of the below comments are addressed.

Thank you for your review, and we agree that this study is exciting to advance the understanding of benthic invertebrate diet. We have addressed your comments below in a stepwise fashion. We are grateful to your attention to detail and have fixed many inconsistencies that you have pointed out with our manuscript. Please note that line numbers below refer to the manuscript with tracked changes for your convenience in locating our changes.

Major Comments

Note that the references as cited in text and in the reference section don’t match the PLOS One style of citation. Please see https://journals.plos.org/plosone/s/submission-guidelines#loc-references

“References are listed at the end of the manuscript and numbered in the order that they appear in the text. In the text, cite the reference number in square brackets (e.g., “We used the techniques developed by our colleagues [19] to analyze the data”). PLOS uses the numbered citation (citation-sequence) method and first six authors, et al.”

Thank you for noticing this oversight, we have now reformatted the references to the correct PLOS ONE style.

Adding a figure of a map where station locations are shown would be helpful in addition to the listing of the latitudes and longitudes. It will illustrate the difference between the sheltered and open sites better as well and where they are located in space.

A map of the stations has been added to the supplemental information of the manuscript, and can be found under “S1 Figure.”

Why did you pool stations for collection? Why did the northern cluster have more stations than the southern cluster? 24 km between the northern stations seems like it could make a difference in physical characteristics like the phytoplankton blooms in the region or the physical forcings you discuss? Is that possible? Please elaborate more on why you chose to pool the stations together.

For this experiment, we were interested in general trends in Macoma balthica diet when it came to these two phytoplankton species, and most importantly, proof of concept that DNA detection methods could be used in dietary studies of this detritivore clam, which had not been done before. Thus, we sampled clams from more than one station in both areas to offset local irregularities (for example, in deposition vs. transverse bottoms). The northern stations were sampled during another sampling campaign, where it was possible to take more stations, but also there is a lower density of M. balthica in the northern stations, so we needed more individuals. For the distance between stations, 24 km will not make a relevant difference in phytoplankton blooms, as sedimentation drives the input of these blooms to the sediment, and this is influenced by physical characteristics, hence why we sampled from several stations for each area. There are absolutely local physical characteristics (such as depth, bottom type) that influence individual M. balthica feeding, but we were interested in population level trends, so pooling individuals over several stations removes the individual differences and focuses on the regional physical characteristics (such as currents and waves). We also wish to point out that we selected areas that were physically different in some ways (such as physical exposure), but similar in many ways (such as depth, salinity and temperature). It is possible that other physical characteristics influence diet of M. balthica on the population level more than physical forcing, but this is what we investigated here. A brief synopsis of this reasoning has been added to lines 139-143 and reads “Briefly, multiple stations were pooled within each area to counteract any local irregularities in bottom composition or sedimentation rate while still preserving the regional trends in physical characteristics to ensure that we were investigating population level effects, and not individual diet. M. balthica density is lower in the northern stations, and thus more stations were needed to obtain enough individuals.”

Where were the samples held/experiments conducted after collection? An incubator at the university? Or at a different location.

The samples were stored and the experiment performed at Stockholm University, in a temperature-constant room. This information has been added to lines 134, 146-147, and 167.

Line 176-177 How did you arrive at those concentrations of each of the phytoplankton species to use? I would like to see the same justification for those values as you use for the number of clams you put into each mesocosm (which was great). Additionally, is there a more recent estimate of the number of individuals per meter squared in addition to the 1976 reference? Populations can change pretty drastically over time, so are the values the same ~40 years later?

The phytoplankton densities were roughly (because our focal species were not investigated) based on sedimentation rates found in Höglander et al. 2004, which has been added to the article in lines 172-173. We have also added Broman et al. 2019 as a more recent reference for M. balthica densities.

Why are the mesocosms and sediment heights different sizes in the fed vs unfed (lines 178 and 182)?

This was due to limitations on mesocosms of the same size; we only had four mesocosms in each size available, and we chose to feed the clams in the smaller mesocosms to concentrate the food, and then give the clams more space for the longer experiment. The surface area was similar between the two sizes, with the larger mesocosms being only 1.7 times larger than the smaller mesocosms.

Line 151-153 How often was water replenished? Was this done manually or through a flow through system? What did maintaining this for the 6 months before the experiments started look like? i.e. daily checks on salinity? Temperature? Water changes? Etc.

Water replenishment was done manually on an as-needed basis. Checks were performed at least twice per week for water level, temperature, and oxygenation, while salinity was checked more infrequently, approximately once a month.

In the discussion, particularly the section “Feeding of the Baltic clam on two phytoplankton species”, I would like to see a little more on why you think cyanobacteria was so present when it is the less nutritious food. I think lines 386-390 is trying to do that, but it is a little confusing to follow. – see note below in minor comments about that particular sentence.

This sentence has been revised and now reads “This incorporation occurs despite cyanobacteria’s lower nutritional status as a food source for benthic and planktonic consumers [60,62]. While N. spumigena blooms have been occurring in the Baltic Sea since its formation [86], they are relatively patchy [87], and buoyant, resulting in a large part of the bloom being consumed in the water column [88]” in lines 379-383. Although cyanobacteria has been considered a low-quality food source, there are many studies that document detritivores (Karlson et al. 2008; Nascimento et al. 2008; Gorokhova 2009; Motwani et al. 2018) and M. balthica (Hedberg et al. 2020) utilizing this allegedly low-quality source. Our understanding is that cyanobacteria are still a source of fresh organic matter, and we chose to focus in this manuscript instead on the factors governing the rate at which M. balthica consume cyanobacteria, rather than why they consume cyanobacteria.

Please be consistent when spelling out the genus of the organisms vs. using the abbreviated start – see specific line comments below under minor comments.

Thank you for your attention to these details. After consulting PLOS ONE, we have updated our species names to follow the guidelines, i.e. full genus and species names are in the title and first mention in the paper, and all subsequent mentions are the abbreviated genus and species.

The figures all appear to be a little bit blurry, please try and fix the resolution.

Thank you, we have updated the figure resolutions.

I would consider adding your results, discussion, and figure on the gut clearing analysis in the main text. I think the arguments you make there help to justify why this method in addition to stable isotope analyses is important, a point you bring up in the introduction of the manuscript.

Thank you for your appreciation of this section, we agree that it is an interesting finding that is important for stable isotope analysis and the future of dietary studies. However, we believe that it will only be of use for a small subsection of our readers, and we do not wish to dilute our main message and findings with the extra noise of this side experiment. We are glad that you found it important!

Minor Comments

Key words – Limecola balthica vs. Macoma balthica in the title. May be a bit confusing, but since they are the same species, perhaps it is to maximize the ability for people to find the paper? Could the key word look like it did in the text Macoma (Limecola) balthica?

Yes, this is to maximize the search terms for the focal species, as the name has only recently been restored to Macoma (see Nielsen 2021 for clarification regarding the name changes).

Line 26, 27, 28 – Can you include the taxonomic citation for the species? For example Macoma calcarea (Gmelin, 1791)

This information has been added to the first reference to M. balthica in the main text (line 78) and reads “Below the photic zone, only a few detritivorous macrofauna species are present; one of the most important is the Baltic clam Macoma (Limecola) balthica (Linnaeus, 1758).”

Line 29 – What is meant by aged sediment? Can you be more specific please?

This phrase has been revised to state “transferred back into their original sediment” to be more specific about the sediment.

Line 65 Add “such as” before “soft”

The suggested change has been made.

Line 68 Change “references” to “reference”

The suggested change has been made.

Line 65/66 Adding a few more details about why they are not readily observable, for example is it because the soft bodied prey decay faster?

The sentence starting in line 59 has been modified to read “They allow for identification of prey species in predators that are not readily observable due to sampling difficulties [16], such as soft bodied prey that cannot be identified in the guts due to faster digestion [17], or cryptic prey species [15].”

Line 80 Remove the extra space after the “(“

The suggested change has been made.

Line 94 Which greenhouse gases?

The cited study focused on methane, which has been added and the sentence starting in line 80 now reads “According to [49], M. balthica is responsible for mineralization of over a third of sedimented phytoplankton in the Baltic Proper, but also increases greenhouse gas (methane) emissions from sediment activity [50].”

Line 94, 95, 108 When you start the sentence with the genus, spell out the full genus – please replace “M.” with Macoma” – there are a few other instances throughout the manuscript of this, please fix and standardize all.

The manuscript has been updated to PLOS ONE guidelines regarding species nomenclature – only mentions of the species in titles or the first mention of the manuscript should be written out fully; all subsequent mentions should be abbreviated genus species. Thank you for your attention to detail of this inconsistency in the previous version of the manuscript.

Line 95 Please list examples of the fish species that eat M. balthica

The example of the European flounder has been added, and line 82 now reads “M. balthica are also important prey of commercially important benthic fish, such as Platichthys flesus [51].”

Line 98 Can you provide a definition for spring in the Baltic, what months does the bloom generally settle? Same when you mention the summer bloom. Do these blooms overlap at all? Or in space and time are they distinct from one another?

Months have been added to the spring and summer bloom information within the article, and lines 85-90 now reads “Most of benthic secondary production growth in the Baltic Sea is coupled to the settling of the spring bloom (typically April in the study area) while relying on detritus for most of the year [49,52–55]. Due to warmer temperatures, a longer growing season, and nutrient enrichment, the cyanobacteria-dominated summer phytoplankton bloom (typically July-August in the study area) has been increasing in magnitude and duration in the Baltic Sea [56,57].” These blooms do not overlap in time, but both occur in the study areas. The spring bloom generally depletes the nitrogen in the surface waters, which brings the end of the spring diatom bloom and a short non-productive period. When the surface waters warm and a thermocline is established, the summer cyanobacteria bloom, which includes nitrogen fixers, are able to establish. 

Line 101 and 103 Are there values to more specifically describe how much one bloom is increasing in magnitude vs how much the other is decreasing? If yes, could you please add to quantify what increasing and decreasing mean in this ecosystem?

Precise quantifications of phytoplankton bloom magnifications are not possible. The amount of increase and decrease of each bloom magnitude depend on the year and physical conditions, and thus, we do not believe that attempting to quantify these changes adds to the objective of this study. However, we have added a reference to Hjerne et al. 2019, which explores the possibility of quantification around climate-driven changes in the spring phytoplankton bloom.

Line 120 Remove the word “which”

The suggested change has been made.

Line 123 Macoma can be abbreviated to M.

The suggested change has been made.

Line 125 Is the Hedberg et al. 2020 citation in reference to just the diatom prey or both the cyanobacteria and the diatom? If it is not for both, please include a reference for cyanobacteria as prey for these clams

The provided reference is for both diatoms and cyanobacteria. Hedberg et al. 2020 performed an experiment where both species used in this study were fed to several Baltic Sea macrofauna species, including M. balthica in varying mixtures and stable isotopes were used to determine feeding uptake and preference between the two phytoplankton species.

Line 126 and 127 You can abbreviate the genus name of the phytoplankton species and the clam after using the genus name fully one as you do in line 125 – when it is not the beginning of the sentence

The suggested change has been made.

Line 116 and 128 Can you please provide some more details about what you mean when you use the term “trophic plasticity”? Is it that they can have flexibility in where they are feeding because of the different feeding strategies and the broad array of prey?

A definition has been added to line 100 and now reads “DNA-based methodologies could help elucidate feeding preferences and individual trophic plasticity (i.e. changes in feeding strategies and food selection).”

Line 135 Can you please elaborate on a few days and provide a more specific window, for example 2-4 days or 3-5 days?

“A few days” has been changed to “3-5 days.”

Lines 138-142 and Lines 144-147 These sentences are a little long and clunky to read, please revise, perhaps into more than one sentence

These sentences have been revised to add all additional information about the stations in parenthesis at the end of the sentence for easier reading. They now read “Benthic sled drags were used to sample in the southern Stockholm archipelago outside Askö marine research station in Sweden on 20 October 2017 (Fig 1; station S; samples pooled from 2 stations with center 58.83 N, 17.55 E, and maximum distance between stations 1.76 km, average 34 m water depth, range 28-39 m)” and “M. balthica were also sampled with benthic sled drags from the northern Stockholm archipelago, outside of Norrtälje, Sweden, on 18 January 2018 (station N; samples pooled from 5 stations with center 59.54 N, 19.43 E, and maximum distance between stations 24 km, average 50 m water depth, range 37-62 m).”

Line 142 Replace “see Fig 1 for timeline” with “Fig 1”

The suggested change has been made.

Line 144 Spell out Macoma for the start of the sentence

We have followed the PLOS ONE guidelines regarding species names, and thus keep this instance as the abbreviated species name.

Line 156 Southern does not need to be capitalized

The suggested change has been made, and the rest of the manuscript checked for similar issues.

Line 239 Please also include the version of R you were using (like you do in line 255) in addition to the version of the package

The R version has been added.

Line 243 Please define ASV at its first use in the manuscript

We have added “amplicon sequence variant (ASV) to line 240.

Line 283 What do you mean difference in DNA- the amount was the same at every time point?

This sentence has been revised and now reads “Additionally, there were no significant differences in phytoplankton DNA concentrations detected between sampling time points for either S. marinoi or N. spumigena (Table 1), indicating no decrease in phytoplankton DNA detected over time.” starting in line 279.

Line 283 Add the word “significant” between “no” and “differences”

The suggested change has been made.

Line 299 In all of the table and figure captions spell out genus names – and keep it consistent, here you spell out the phytoplankton genus names but not Macoma. If you mention the name a second time within the caption you can use the abbreviated start for the genus.

We agree with your guidelines for genus names in figure and table captions; we have written out the genus name the first mention in each caption, and abbreviated subsequent genus name mentions.

Line 313 – You write, significantly higher, please list what the p value was in parentheses behind amounts or point to the specific part of Table 1 you are referencing, it is a little confusing

The statistics have now been added to this sentence, and it reads “However, the southern population contained significantly higher (F1,50 = 23.5, p < 0.001) amounts of N. spumigena 16S rRNA gene fragments in their guts than the northern population (Fig 2B; Table 1).” in lines 310-311.

Line 347 What is the Stress=0.16? It isn’t a complete sentence so should be added to another sentence or more explanation should be added to make it a complete sentence.

Stress is the disagreement between the plot configuration and predicted regression values, and must be reported for NMDS as a statistical value. We have added it as parenthesis at the end of the caption description rather than its own sentence.

Line 386 Define OM the first time you use it

We have written out “organic matter” instead of OM, as this is the only occasion of use.

Line 386-391 This sentence is hard to follow please revise

This sentence has been revised, splitting it into two sentences for the reader to more easily follow. It now reads “This incorporation occurs despite cyanobacteria’s lower nutritional status as a food source for benthic and planktonic consumers [60,62]. While N. spumigena blooms have been occurring in the Baltic Sea since its formation [86], they are relatively patchy [87], and buoyant, resulting in a large part of the bloom being consumed in the water column [88]” in lines 379-383.

Line 391 This should be the start of a new paragraph – if it is then indenting paragraphs throughout the manuscript would make this clearer

We believe this sentence to be the continuation of the idea started in the previous sentence – that cyanobacteria consumption depends on physical characteristics of the environment. Thus, we respectfully keep lines 369-405 as one paragraph.

Line 394 Organic matter is spelled out here where earlier you used OM, please be consistent, either spell it out every time or define it and then use OM throughout the rest of the discussion

Thank you for pointing this out, we have changed OM from line 379 to organic matter to be consistent.

Line 408-411 What is meant here by phytoplankton fed? I think I understand this sentence, but it gets a little confusing, can you add some details and specifics please.

This sentence has been revised for clarity and now reads “Interestingly, N. spumigena and S. marinoi were also detected in all metabarcoding samples, but were not among the top phytoplankton in terms of relative abundance in the DNA metabarcoding dataset.”

Line 411 After communities add “of phytoplankton”

“Phytoplankton” has been added before “communities” to read “phytoplankton communities.”

Line 428 What kind of intermediate organisms might M. balthica be consuming, can you provide some examples? And are those animals feeding on the species you found inside the clams?

We have added “such as meiofauna” to the first mention of intermediate organisms in line 427. M. balthica have been documented to consume meiofauna (Ólafsson et al. 1993), and meiofauna are known to consume phytoplankton (Nascimento et al. 2008, 2009).

Supplementary Material

Line 4 Add “to” between “macrofauna” and “purge”

The suggested change has been made.

---

## [Decision Letter · Decision Letter 1]

19 Sep 2022

PONE-D-22-11031R1Molecular diet analysis enables detection of diatom and cyanobacteria DNA in the gut of *Macoma balthica*PLOS ONE

Dear Dr. Garrison,

Thank you for submitting your manuscript to PLOS ONE. I have received feedback from the two reviewers on the revised version of the paper, and Reviewer #1 has made some additional suggestions that I think should improve the manuscript and I ask that you consider those suggestions where it is practical. Consequently, I invite you to submit a revised version of the manuscript that addresses the minor revisions that the reviewer recommends. I realize that there are questions that might be challenging to address with the data in hand, but I expect to be able to recommend acceptance of the paper once these points the reviewer makes have been acknowledged and the constraints on using the approach you and your colleagues have undertaken are clearly spelled out. I thank you again for considering PLOS One to publish your results and look forward to receiving a second revised version of the paper. I am currently on a research cruise and have some limits to my email access for the next couple weeks, so please be patient with my response time if you are able to return the manuscript in that time frame.

Formally, we are requesting that you please submit your revised manuscript by Nov 03 2022 11:59PM. If you will need more time than this to complete your revisions, please reply to this message or contact the journal office at plosone@plos.org. Please include the following items when submitting your revised manuscript:A rebuttal letter that responds to each point raised by the academic editor and reviewer(s). You should upload this letter as a separate file labeled 'Response to Reviewers'.A marked-up copy of your manuscript that highlights changes made to the original version. You should upload this as a separate file labeled 'Revised Manuscript with Track Changes'.An unmarked version of your revised paper without tracked changes. You should upload this as a separate file labeled 'Manuscript'.If applicable, we recommend that you deposit your laboratory protocols in protocols.io to enhance the reproducibility of your results. Protocols.io assigns your protocol its own identifier (DOI) so that it can be cited independently in the future. For instructions see: https://journals.plos.org/plosone/s/submission-guidelines#loc-laboratory-protocols. Additionally, PLOS ONE offers an option for publishing peer-reviewed Lab Protocol articles, which describe protocols hosted on protocols.io. Read more information on sharing protocols at https://plos.org/protocols?utm_medium=editorial-email&utm_source=authorletters&utm_campaign=protocols.

We look forward to receiving your revised manuscript.

Kind regards,

Lee W Cooper, Ph.D.

Section Editor

PLOS ONE

Journal Requirements:

Reviewers' comments:

Reviewer's Responses to Questions

**Comments to the Author**

1. If the authors have adequately addressed your comments raised in a previous round of review and you feel that this manuscript is now acceptable for publication, you may indicate that here to bypass the “Comments to the Author” section, enter your conflict of interest statement in the “Confidential to Editor” section, and submit your "Accept" recommendation.

Reviewer #1: (No Response)

Reviewer #2: All comments have been addressed

2. Is the manuscript technically sound, and do the data support the conclusions?

Reviewer #1: Partly

Reviewer #2: Yes

3. Has the statistical analysis been performed appropriately and rigorously? 

Reviewer #1: Yes

Reviewer #2: Yes

4. Have the authors made all data underlying the findings in their manuscript fully available?

Reviewer #1: Yes

Reviewer #2: Yes

5. Is the manuscript presented in an intelligible fashion and written in standard English?

Reviewer #1: Yes

Reviewer #2: Yes

6. Review Comments to the Author

Reviewer #1: This manuscript is much improved, and the inferences presented more reasonable. The paper is still written to be mainly about methodology. However, given that the experiments cannot indicate the duration of detectability of DNA in bivalve guts after consumption, the main methodological conclusion is simply that DNA can be detected in the guts, which was not really questionable beforehand. Nevertheless, your results have important implications for ecology and the body of work on deposit-feeder food sources that could be emphasized more clearly.

In this study, DNA in the clams’ guts before the experiment indicated that, even after 9 months without adding phytoplankton, Macoma were still consuming phytoplankton or at least bacteria that had consumed phytoplankton-derived material. When phytoplankton of particular taxa were then added in substantial amounts to the sediments, there was no increase in relative occurrence of the DNA of those taxa in the clams’ guts from before additions to the end of the experiment (1 week), and these added taxa remained a minor component of gut contents. If that assessment of your results is inaccurate, please state the results more simply and clearly for the many readers on this topic who have not used these methods themselves.

Although not what the experiment set out to investigate, from an ecological perspective the major conclusions appear to be that

(1) The DNA of phytoplankton can remain available in the sediments in ingestable form for at least 6 months after cessation of any additions. Whether this DNA is contained within persistent phytoplankton cells or bacteria that consumed phytoplankton-derived material is unclear.

(2) The levels of phytoplankton-derived material persisting after 6 months can sustain deposit-feeding Macoma.

(3) Pulsed additions of fresh phytoplankton do not alter the clams’ diet within 7 days. This finding might indicate that the clams do not respond within a week to the availability of added material, or that added material requires a period of processing via bacteria or passing through the guts of deposit-feeders before becoming viable foods.

Conclusion #3 is curious, as previous studies have shown that even if deposit-feeders do not assimilate the fresh material, they do in fact ingest it, so the DNA from the fresh material should have been detectable on the clams’ guts.

1. L 35-36. If I understand your results correctly, this statement is not justified. The relative occurrence of neither N. marinois or N. spumigena increased in the clams’ guts from before to after feeding, so there is no confirmation that DNA from the added material per se was still present in the guts at the end of the week-long experiment. DNA from these taxa was present in the sediments and already being consumed in similar amounts before the addition, so we don’t known that the clams were ingesting the added material as opposed to what was already there. If your data do show that they were clearly using newly added material, please point out those data and make the point more plainly in the Results to refute the alternative possibility.

2. L 58. Is reference [14] a thesis? If so, say so in the reference list. If not, without more information it is not apparent how to obtain this document. Only references that are generally available should be cited. Is there perhaps a doi number?

3. L 116. “physical forcing influences on their feeding strategy”. The meaning of this phrase is unclear. Just say "differing availability of the two phytoplankton species". You don’t know whether bloom dynamics, current patterns, or water-column processes (buoyancy, predation by zooplankton) determine their relative availability, so just stick with the proximate factor that the animals will actually respond to.

4. L 118-119. This phrasing begs the question of why the southern population would exhibit higher deposit-feeding, as this prediction has no basis in preceding text. Deposit-feeding is not an inherently more profitable feeding mode than filter-feeding. The difference must result from variations in food availability accessible by the different feeding modes. I suggest the wording "... southern population, with longer open-water period and settlement of greater phytoplankton biomass (reference), would have higher feeding rates."

5. L 172-173. Please state explicitly that the -24 clams were not exposed to the freshly amended sediments before being sampled, if that was the case. Bivalves can respond almost immediately to food additions (within 20 min), although they do not always do so depending on prior conditions.

Also, if you have the data, please state the range of shell lengths of clams added to the mesocosms.

6. L 273-274. Although there was no decrease, there was also no increase at the beginning or throughout the experiment, which is perhaps your most interesting finding. The clams were already consuming DNA of the experimental phytoplankton taxa at comparable rates before the experimental additions.

7. L 275. Another striking aspect is that the variation among replicates in consumption of the added diatom taxon increased greatly toward the end of the experiment in both regions. Although response times certainly vary, some experiments have revealed that deposit-feeders can take at least a week to react to additions of phytoplankton (e.g. Deep-Sea Res I 55:1503, 2008). It is possible that the increased variation in consumption of S. marinoi toward the end of your experiments indicated that some individuals were just starting to respond to the food addition from a week before.

8. L 283. Please increase the font sizes of the axis labels in Fig. 4, as they are currently too small to be read easily.

9. L 301-304. The higher level of N. spumigena DNA in the guts of the southern population was evident before any experimental addition (after 6 mo without additions of food), and the rate of feeding on N. spumagena was unaffected by additions. It appears that the N spumagena DNA was already available at a higher level in the sediments from the southern population before the experiments began.

10. L 353-354. If you did not see an increase after vs. before phytoplankton additions, why would you expect to see a decrease? The clams were already subsisting on phytoplankton (or bacteria that had consumed phytoplankton material) before the experiment started, and continued to feed on the same phytoplankton-derived material that was already present in the sediments after feeding. There was no detectable change in diet after phytoplankton additions.

11. L 419-420. Although Macoma can affect meiofauna via trophic competition, I am unaware of studies showing actual consumption of meiofauna by Macoma. I would use another organism as an example here (juvenile fish commonly eat meiofauna), and then stick with Macoma consuming bacteria in subsequent text.

12. L 434-435. Replace "by accident" with "incidentally"

13. L 442. The phrase "high trophic plasticity from potentially variations of feeding strategies" is awkward and has unclear meaning. The most logical explanation is that cyanobacterial material was simply more available in the sediments from the southern region. With planktonic larvae, Macoma from sampling regions so close together are likely all from the same population genetically, and they are not necessarily varying their feeding mode. For Macoma in the southern sediments that consumed greater amounts of cyanobacterial material before the addition experiments, did you notice that they had their siphons extended more often? If not, it appears that feeding mode did not explain the difference in consumption of cyanobacteria.

More generally, I would avoid language implying that Macoma from the two sampling regions were in fact different "populations". The sampling regions are quite close together, and Macoma has pelagic larvae that could easily disperse between these areas. I doubt that there are genetic differences between them, and a number of readers think of “populations” as being geographically or genetically differentiated. I would avoid reference to the "two populations" and rely on the term "regions".

14. L 445. From the perspective of “ecological” (food web) models, the term “feeding rate" here just means diet, as your "feeding rate" data simply indicate relative proportions of different foods consumed. The DNA data do not indicate absolute amounts (in mass) of different foods consumed per unit time, which is the way “feeding rate” in trophic analyses is defined. Thus, in relating your data to ecological models, the terms "diet" and "feeding rate" are redundant. In this paper you are trying to appeal to people concerned with trophic analyses, so I would come up with a different term than “feeding rate” to avoid confusion.

Reviewer #2: (No Response)

7. PLOS authors have the option to publish the peer review history of their article (what does this mean?). If published, this will include your full peer review and any attached files.

Reviewer #1: No

Reviewer #2: No

---

## [Author Response · Author response to Decision Letter 1]

3 Nov 2022

Thank you for submitting your manuscript to PLOS ONE. I have received feedback from the two reviewers on the revised version of the paper, and Reviewer #1 has made some additional suggestions that I think should improve the manuscript and I ask that you consider those suggestions where it is practical. Consequently, I invite you to submit a revised version of the manuscript that addresses the minor revisions that the reviewer recommends. I realize that there are questions that might be challenging to address with the data in hand, but I expect to be able to recommend acceptance of the paper once these points the reviewer makes have been acknowledged and the constraints on using the approach you and your colleagues have undertaken are clearly spelled out. I thank you again for considering PLOS One to publish your results and look forward to receiving a second revised version of the paper. I am currently on a research cruise and have some limits to my email access for the next couple weeks, so please be patient with my response time if you are able to return the manuscript in that time frame.

Dear Dr. Cooper,

Thank you for the opportunity to submit our revised manuscript. We are very glad to more clearly communicate the constraints as well as the strengths of our data, and are appreciative of reviewer 1’s comments. The reviewer has helped point out the less unsupported claims that we have made with our data, as well as highlighting some interesting and unexpected results of our study that will appeal to a broader audience. We have stepwise addressed all the reviewer’s comments and made changes to the manuscript, as well as supplementary information. Please find replies to the reviewer comments, as well as manuscript text changes and referenced line numbers in the tracked changes version of the manuscript.

We are very excited about the prospect of publishing our manuscript with PLOS ONE.

Sincerely,

Julie Garrison and co-authors

 

Reviewer #1: This manuscript is much improved, and the inferences presented more reasonable. The paper is still written to be mainly about methodology. However, given that the experiments cannot indicate the duration of detectability of DNA in bivalve guts after consumption, the main methodological conclusion is simply that DNA can be detected in the guts, which was not really questionable beforehand. Nevertheless, your results have important implications for ecology and the body of work on deposit-feeder food sources that could be emphasized more clearly.

Thank you for reviewing our manuscript again. We appreciate the time that you have taken, and agree that our manuscript focuses on methodology and an aim that we could not meet. We have revised the manuscript to focus more broadly on deposit-feeder diet, specifically altering the end of the abstract to focus on the DNA metabarcoding diet results (lines 39-43), the first paragraph of the discussion to focus on your conclusion 3 from the next comment (lines 366-371), and recommending future studies to fill the gaps left by our study (lines 433-435 and 438-440). We hope that you find these changes adequate to redirect our manuscript from a methodological study to an exploratory one.

Lines 39-43: “However, DNA metabarcoding of the 23S rRNA phytoplankton gene found in the clam guts showed that added food (i.e. N. spumigena and S. marinoi) did not make up a majority of the detected diet. Our results suggest that these detritivorous clams therefore do not react as quickly as previously thought to fresh organic matter inputs, with other phytoplankton than large diatoms and cyanobacteria constituting the majority of their diet.”

Lines 366-371: “This could be a result of too little time to react to the fresh organic matter, which was 24 hours in our study, and perhaps varied access time to fresh food should be evaluated in future experiments. Additionally, we found that phytoplankton DNA in sediments can remain viable and ingested by benthic detritivores for at least six months following organic matter input, which can have important implications for the understanding of benthic detritivore feeding ecology.”

Lines 433-435: “This finding furthers our hypothesis that the M. balthica from different stations have different trophic strategies, but could also simply be an indication of different phytoplankton in the sediment, and would require analysis of the phytoplankton DNA present in the sediment before the additions of organic matter made in our experiment to make accurate conclusions.”

Lines 438-440: “Interestingly, our study indicates that phytoplankton DNA remains available in the sediment in ingestible form for at least six months after fresh organic matter addition, and is consumed by benthic detritivores.”

In this study, DNA in the clams’ guts before the experiment indicated that, even after 9 months without adding phytoplankton, Macoma were still consuming phytoplankton or at least bacteria that had consumed phytoplankton-derived material. When phytoplankton of particular taxa were then added in substantial amounts to the sediments, there was no increase in relative occurrence of the DNA of those taxa in the clams’ guts from before additions to the end of the experiment (1 week), and these added taxa remained a minor component of gut contents. If that assessment of your results is inaccurate, please state the results more simply and clearly for the many readers on this topic who have not used these methods themselves.

Although not what the experiment set out to investigate, from an ecological perspective the major conclusions appear to be that

(1) The DNA of phytoplankton can remain available in the sediments in ingestable form for at least 6 months after cessation of any additions. Whether this DNA is contained within persistent phytoplankton cells or bacteria that consumed phytoplankton-derived material is unclear.

(2) The levels of phytoplankton-derived material persisting after 6 months can sustain deposit-feeding Macoma.

(3) Pulsed additions of fresh phytoplankton do not alter the clams’ diet within 7 days. This finding might indicate that the clams do not respond within a week to the availability of added material, or that added material requires a period of processing via bacteria or passing through the guts of deposit-feeders before becoming viable foods.

Conclusion #3 is curious, as previous studies have shown that even if deposit-feeders do not assimilate the fresh material, they do in fact ingest it, so the DNA from the fresh material should have been detectable on the clams’ guts.

We have modified the manuscript to better reflect the three conclusions underlined by the reviewer. We agree without exception with your first two conclusions presented here from our work, and we have highlighted these conclusions in the discussion of our manuscript (lines 366-371, 433-435, and 438-440, all mentioned above). We also agree mostly with your third conclusion, with the provision that the clams only had access to the fresh organic matter for 24 hours, giving support to your hypothesis that Macoma balthica requires a period of processing before it can process freshly deposited phytodetritus as food. We have added this information to the discussion section in lines 366-371. As discussed in multiple places within the manuscript, this is the first study that we are aware of that investigates benthic detritivore diet by molecular methods, and multiple improvements can be made in future work. We are happy to make recommendations for future work on this topic, and hopeful that future researchers will be able to learn from our mistakes. We also agree that our results are interesting and open up a number of questions, which hopefully will be evaluated in future work. While our data did not support our original aims, we have found support for these conclusions that are perhaps more broadly interesting than the study that we set out to perform.

All three conclusions have been highlighted in our conclusions section, which now reads “Our results suggest that the feeding of two regional groups of the Baltic clam on a diatom species do not differ, but clams from the more sheltered site fed significantly more on a cyanobacteria species, possibly indicating higher availability of cyanobacterial material or higher trophic plasticity that results in potential variations of feeding strategies. However, these fresh organic matter additions did not make up the majority of the clams’ diet. Our experimental data suggests a significant proportion of the diet is made up of picocyanobacteria and unicellular green algae, which were not added freshly during the experiment. This indicates that phytoplankton DNA can remain in sediments for at least six months, and that these phytoplankton can sustain deposit-feeding Macoma balthica” in lines 470-479.

1. L 35-36. If I understand your results correctly, this statement is not justified. The relative occurrence of neither N. marinois or N. spumigena increased in the clams’ guts from before to after feeding, so there is no confirmation that DNA from the added material per se was still present in the guts at the end of the week-long experiment. DNA from these taxa was present in the sediments and already being consumed in similar amounts before the addition, so we don’t known that the clams were ingesting the added material as opposed to what was already there. If your data do show that they were clearly using newly added material, please point out those data and make the point more plainly in the Results to refute the alternative possibility.

This sentence has been modified to read “Interestingly, the cyanobacteria and diatom DNA fragments were still detectable by qPCR in the guts of M. balthica one week after removal from its food source. However, DNA metabarcoding of the 23S rRNA phytoplankton gene found in the clam guts showed that added food (i.e. N. spumigena and S. marinoi) did not make up a majority of the detected diet. Our results suggest that these detritivorous clams therefore do not react as quickly as previously thought to fresh organic matter inputs, with other phytoplankton than large diatoms and cyanobacteria constituting the majority of their diet” in lines 37-43. We agree that we cannot be certain that the clams are indeed consuming the added fresh phytoplankton and not aged material from the original sediments (but see our response to comment 10), so we have modified to point out more of our findings from the DNA metabarcoding results and highlight your conclusion 3 from the previous comment.

2. L 58. Is reference [14] a thesis? If so, say so in the reference list. If not, without more information it is not apparent how to obtain this document. Only references that are generally available should be cited. Is there perhaps a doi number?

Yes, reference [14] is a PhD thesis, and has been updated in the references to make it clear. This thesis is available, and a link has been added to aid readers to find it.

14. Hedberg P. Responses of benthic-pelagic coupling to environmental change. PhD thesis, Stockholm University. 2021. Available: http://su.diva-portal.org/smash/record.jsf?pid=diva2%3A1594570&dswid=-3644

3. L 116. “physical forcing influences on their feeding strategy”. The meaning of this phrase is unclear. Just say "differing availability of the two phytoplankton species". You don’t know whether bloom dynamics, current patterns, or water-column processes (buoyancy, predation by zooplankton) determine their relative availability, so just stick with the proximate factor that the animals will actually respond to.

We agree with the reviewer’s assessment that we cannot be sure that differences in feeding between the two groups of clams would be due to physical forcing. We have revised this sentence to read “ii) if feeding by the two M. balthica regional groups on each of the phytoplankton species differs, which would likely be due to trophic plasticity and differing availability of phytoplankton on their feeding strategy” in lines 122-125.

4. L 118-119. This phrasing begs the question of why the southern population would exhibit higher deposit-feeding, as this prediction has no basis in preceding text. Deposit-feeding is not an inherently more profitable feeding mode than filter-feeding. The difference must result from variations in food availability accessible by the different feeding modes. I suggest the wording "... southern population, with longer open-water period and settlement of greater phytoplankton biomass (reference), would have higher feeding rates."

We agree that this statement was not adequately supported. We have revised the text to better connect reference [66] with this statement, where clams in sheltered habitats were found to spend more time deposit feeding than clams from exposed habitats. Since the phytoplankton was mixed into the sediment in the experiment, the sheltered southern clams should have higher feeding magnitudes due to their preference for deposit feeding. The sentence now reads “We predicted that both phytoplankton species would be consumed by all clams, but that S. marinoi would be consumed at a higher magnitude due to the higher nutritional value, and that the sheltered southern clams would have higher phytoplankton consumption due to higher deposit feeding, where the phytoplankton in this experiment could be found” in lines 125-129.

5. L 172-173. Please state explicitly that the -24 clams were not exposed to the freshly amended sediments before being sampled, if that was the case. Bivalves can respond almost immediately to food additions (within 20 min), although they do not always do so depending on prior conditions.

Also, if you have the data, please state the range of shell lengths of clams added to the mesocosms.

Yes, your assessment is correct that -24 h clams were not exposed to fresh phytoplankton, and we have tried to make this more clear in lines 184-185, but would also like to point out that this information was already available in lines 198 and figure 1. We agree that the literature indicates that bivalves can respond almost immediately to fresh food additions, and have included the suggestion to take samples sooner after feeding than 24 hours in our conclusions (lines 366-371; mentioned in previous replies). We did not measure precisely the shell lengths of individual clams, but have added the range 10 to 20 mm to our methods (lines 200-202) and emphasize the wet weight available in Table S1. The text now reads “For M. balthica wet weight, please see S1 Table. While individual shell lengths were not measured, all clams were between 10-20 mm to reduce variation by life stage and particle size processing limitations.”

6. L 273-274. Although there was no decrease, there was also no increase at the beginning or throughout the experiment, which is perhaps your most interesting finding. The clams were already consuming DNA of the experimental phytoplankton taxa at comparable rates before the experimental additions.

Yes, you are correct. We have altered this sentence to highlight the lack of increase as well, and it now reads “Additionally, there were no significant differences in phytoplankton DNA concentrations detected between sampling time points for either S. marinoi or N. spumigena (Table 1), indicating no significant increase or decrease in phytoplankton DNA detected over time” in lines 286-288. We have also highlighted this finding in our discussion section, as mentioned in a previous response.

7. L 275. Another striking aspect is that the variation among replicates in consumption of the added diatom taxon increased greatly toward the end of the experiment in both regions. Although response times certainly vary, some experiments have revealed that deposit-feeders can take at least a week to react to additions of phytoplankton (e.g. Deep-Sea Res I 55:1503, 2008). It is possible that the increased variation in consumption of S. marinoi toward the end of your experiments indicated that some individuals were just starting to respond to the food addition from a week before.

Yes, we also have noticed the large variation between replicates, but since the difference between timepoints in ingestion of diatoms was not statistically significant we cannot be certain that it is actual biological variation and not methodological, or a result of only three replicates per time point and treatment. We have suggestions to future research both to extend the time that molecular monitoring of diet occurs in line 374, and to increase numbers of replication in line 383. The higher replication would allow future experiments to tease out if the large variation is due to individual diet differences or limitations of the methods, and longer experiment times will hopefully allow for a decline in detected amended phytoplankton. 

8. L 283. Please increase the font sizes of the axis labels in Fig. 4, as they are currently too small to be read easily.

The font sizes in figure 4 have been increased.

9. L 301-304. The higher level of N. spumigena DNA in the guts of the southern population was evident before any experimental addition (after 6 mo without additions of food), and the rate of feeding on N. spumagena was unaffected by additions. It appears that the N spumagena DNA was already available at a higher level in the sediments from the southern population before the experiments began.

Yes, the southern clams started with higher N. spumigena than the northern clams, even without additions of fresh material. The southern clams were even without new food sources for 3 months longer than the northern clams, indicating that the clams were possibly actively avoiding N. spumigena in the sediment. However, we do see an increase from the baseline -24 value after 48 and 72 hours in the southern clams. In contrast, the northern clams demonstrated an immediate increase already at 0 h from the baseline, and a sharp increase after 48 and 144 h.

In accordance with the results, we have added to the results and discussion sections cautioning against over interpretation of our results. The results reads “However, the southern clams contained significantly higher (F1,50 = 23.5, p < 0.001) amounts of N. spumigena 16S rRNA gene fragments in their guts than the northern clams (Fig 2B; Table 1), but the southern clams showed already higher baseline values at -24 h (Fig 2B)” in lines 317-320. The discussion section now reads “Additionally, sediment samples from before, during, and after the experiment would be required to confirm the targeted phytoplankton DNA levels in the sediment. Indeed, the baseline of N. spumigena in the southern clams was higher than the northern clams before fresh phytoplankton were added, and these results must be taken with that consideration” in lines 414-418.

10. L 353-354. If you did not see an increase after vs. before phytoplankton additions, why would you expect to see a decrease? The clams were already subsisting on phytoplankton (or bacteria that had consumed phytoplankton material) before the experiment started, and continued to feed on the same phytoplankton-derived material that was already present in the sediments after feeding. There was no detectable change in diet after phytoplankton additions.

This section in lines 372-384 of the discussion serves to explain some potential reasons why we did not see an increase after feeding or a linear decrease as seen in previous studies, i.e. too few sampling points, sampling points not close enough to the feeding, or far enough from the feeding, or too few replicates or any other potential reasons. Out of our two groups of M. balthica and two phytoplankton species, we did see two out of four treatments increase (although statistically insignificant) from the baseline following feeding: northern clams feeding on S. marinoi and northern clams feeding on N. spumigena. When also including the emptied gut treatments presented in the supplementary material, we find one additional treatment where feeding increased phytoplankton DNA detection from the baseline: southern clams feeding on N. spumigena with emptied guts. We cannot be sure why all of our treatments didn’t increase from the baseline following feeding, and this should be investigated in future studies. Additionally, of these 3 treatments which showed an increase following feeding, none of them showed a linear decrease from the initial increase.

We have added the lack of increase following feeding to our first sentence of this paragraph, which now reads “Unlike previous studies that looked at the prevalence of prey DNA in e.g. insect and spider consumers [25,29,33,35–38], we found no consistent increase in phytoplankton detection or linear decrease in the guts of the clam M. balthica following feeding” in lines 372-374.

11. L 419-420. Although Macoma can affect meiofauna via trophic competition, I am unaware of studies showing actual consumption of meiofauna by Macoma. I would use another organism as an example here (juvenile fish commonly eat meiofauna), and then stick with Macoma consuming bacteria in subsequent text.

We have added a reference to this sentence to Ólafsson et al. 1993 [52], which shows that M. balthica consumes harpacticoid copepods in line 448. This finding comes from a feeding experiment, where a reduction in number of harpacticoid copepods was observed when sharing a habitat with M. balthica, and thus has limitations, but is widely cited as evidence of M. balthica feeding on meiofauna. We also believe that a study with newer molecular methods would be beneficial.

12. L 434-435. Replace "by accident" with "incidentally"

The suggested change has been made in line 466.

13. L 442. The phrase "high trophic plasticity from potentially variations of feeding strategies" is awkward and has unclear meaning. The most logical explanation is that cyanobacterial material was simply more available in the sediments from the southern region. With planktonic larvae, Macoma from sampling regions so close together are likely all from the same population genetically, and they are not necessarily varying their feeding mode. For Macoma in the southern sediments that consumed greater amounts of cyanobacterial material before the addition experiments, did you notice that they had their siphons extended more often? If not, it appears that feeding mode did not explain the difference in consumption of cyanobacteria.

The phrase has been revised to “Our results suggest that the feeding of two regional groups of the Baltic clam on a diatom species do not differ, but clams from the more sheltered site fed significantly more on a cyanobacteria species, possibly indicating higher availability of cyanobacterial material or higher trophic plasticity that results in potential variations of feeding strategies” in line 470. Unfortunately, no reliable observations of behavior were made during the experiment. We include the word “potential” to highlight that we did not observe this difference in feeding strategy during our experiment, but found it a likely explanation for our results. Of course, additional studies should be conducted to investigate this, and we recommend to observe the feeding strategies, by for example observing time spent with siphons extended and providing food in different methods (mixing into sediment vs. sedimenting from the water column vs. sediment surface).

More generally, I would avoid language implying that Macoma from the two sampling regions were in fact different "populations". The sampling regions are quite close together, and Macoma has pelagic larvae that could easily disperse between these areas. I doubt that there are genetic differences between them, and a number of readers think of “populations” as being geographically or genetically differentiated. I would avoid reference to the "two populations" and rely on the term "regions".

We have revised the manuscript to avoid the term “population” when describing clams from the two regions. Instead, we have followed your suggestion and used “northern clams,” “southern clams,” and “regions” to describe these groups. Changes have been implemented throughout the manuscript and in figures 3 and 4, as well as supplementary information.

14. L 445. From the perspective of “ecological” (food web) models, the term “feeding rate" here just means diet, as your "feeding rate" data simply indicate relative proportions of different foods consumed. The DNA data do not indicate absolute amounts (in mass) of different foods consumed per unit time, which is the way “feeding rate” in trophic analyses is defined. Thus, in relating your data to ecological models, the terms "diet" and "feeding rate" are redundant. In this paper you are trying to appeal to people concerned with trophic analyses, so I would come up with a different term than “feeding rate” to avoid confusion.

We agree that “feeding rate” is not the most accurate term that could be used here, and have made changes throughout the manuscript to instead refer to this detection of food over time as simply “feeding” or “feeding magnitude.”

---

## [Editor Report · Decision Letter 2]

9 Nov 2022

Molecular diet analysis enables detection of diatom and cyanobacteria DNA in the gut of *Macoma balthica*

PONE-D-22-11031R2

Dear Ms. Garrison,

Thank you for working to address the second set of reviewer comments from Reviewer #1. I have read through the revised manuscript and response to reviewer comments and I am pleased to let you know that I think you have successfully addressed all of the remaining concerns and that your manuscript is scientifically suitable for publication. It will be formally accepted for publication once it meets all outstanding technical requirements. Thank you for working with the reviewers and I trust that the peer review process with PLOS has been positive in your experience and has improved the communication of your interesting findings for the feeding patterns of Baltic bivalves.

Kind regards,

Lee W Cooper, Ph.D.

Section Editor

PLOS ONE
---

## [Editor Report · Acceptance letter]

15 Nov 2022

PONE-D-22-11031R2 

Molecular diet analysis enables detection of diatom and cyanobacteria DNA in the gut of *Macoma balthica*

Dear Dr. Garrison:

I'm pleased to inform you that your manuscript has been deemed suitable for publication in PLOS ONE. Congratulations! Your manuscript is now with our production department. 

Kind regards, 

on behalf of

Dr. Lee W Cooper 

Section Editor

PLOS ONE